# DBMSOLVER: FAST DIFFUSION BRIDGE SAMPLING FOR HIGH-QUALITY IMAGE-TO-IMAGE TRANSLATION

## ABSTRACT

Diffusion-based approaches for image-to-image (I2I) translation have garnered significant attention due to their ability to generate high-fidelity images and scalability to large-scale datasets. However, state-of-the-art Diffusion Bridge Models (DBMs), which utilize diffusion bridges to interpolate between two images $\mathbf{x}_0$ and $\mathbf{x}_T$, are severely hampered by their slow sampling process, often requiring dozens to hundreds of function evaluations. To address this computational burden, we introduce **DBMSolver**, a novel, training-free sampler specifically designed for DBMs. DBMSolver leverages the inherent semi-linear structure of the underlying diffusion equations in DBMs and employs advanced exponential integrators to accelerate the sampling process. This approach not only reduces the number of evaluations but also enhances image quality for I2I Translation tasks. Our experiments demonstrate that DBMSolver outperforms prior methods across multiple datasets and resolutions, significantly improving visual quality and reducing computational overhead. DBMSolver improves the scalability of diffusion-based I2I Translation by bridging the gap between theoretical elegance and real-world applicability.

## 1 INTRODUCTION

Image-to-Image (I2I) Translation is a generative modeling paradigm that learns to map an input image to a target output. It encompasses tasks like image restoration, grayscale colorization, and inpainting of occluded or corrupted regions, as well as style transfer and semantic reinterpretation via cross-domain synthesis (Sxela, 2021; Isola et al., 2017; Goodfellow et al., 2014).

Recent diffusion-based works, as alternatives to traditional generative approaches such as GANs (Zhu et al., 2017; Karras et al., 2020), have brought significant advances in the synthesis of high-fidelity images (Saharia et al., 2022a; Liu et al., 2023; Kawar et al., 2022). Among them, Zhou et al. (2023) proposed Diffusion Bridge Models (DBMs), which are capable of performing I2I Translation by establishing a *diffusion bridge* that facilitates the translation from one arbitrary image distribution

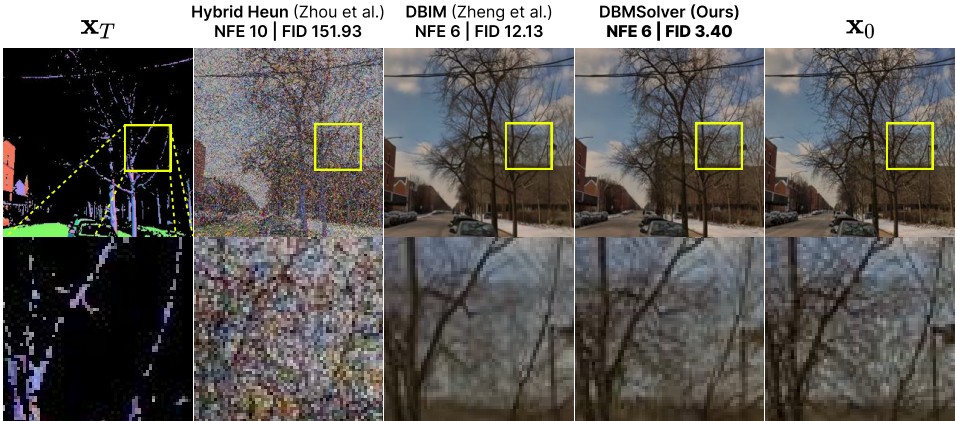

Figure 1: Few-step image synthesis (6 NFEs ↓) with high-quality generated details (3.40 FID ↓).

to another. While DBMs offer a theoretically elegant diffusion framework for I2I Translation, generating high-quality images using such diffusion-based models remains computationally intensive as it requires numerous costly model evaluations (NFEs)– which forms the core of our contribution.

## 1.1 DBMSOLVER: OUR SOLUTION AND KEY CONTRIBUTIONS

We introduce *DBMSolver*: a training-free, highly efficient solver specifically designed to accelerate the DBM-based sampling process. Previous research on improving the sampling speeds explored either model distillation (He et al., 2024; Xie et al., 2024; Gushchin et al., 2025), fine-tuning (Geng et al., 2024; He et al., 2024), or re-training of an entire neural network (Zhou et al., 2025). In contrast, DBMSolver is a drop-in replacement for existing DBM sampling methods, avoiding the need for any architectural changes or extra training, thus enabling broad compatibility and immediate benefits.

We devise DBMSolver by rigorously analyzing the underlying Stochastic and Ordinary Differential Equations (SDE and ODE) governing DBMs' reverse-time diffusion process (Equations 2 and 5). Specifically, in Section 3.1, we identify the inherent semi-linear structure of the SDE, and leverage the Exponential Integrators (EI) (Hochbruck & Ostermann, 2010) method to derive its exact solution. Next, in Section 3.2, we analyze the ODE to show that it has a semi-linear structure as well, allowing us to derive a $2^{\text{nd}}$-order exact solution. Finally, by combining these two solutions, we devise a sampling procedure that drastically reduces the required NFEs while enhancing the image quality, as showcased in Figure 1 and Section 3.3. The key contributions of this work are as follows:

- **Novel, Training-Free Sampler**: DBMSolver significantly accelerates the sampling speed of Diffusion Bridge Models without requiring any additional training or fine-tuning. It works for both, conditional and unconditional I2I Translation.

- **Principled Theoretical Foundation**: We implement DBMSolver with exact solutions of the SDE and ODE governing DBMs, grounding it on the principled diffusion bridge theory.

- **State-of-the-art Performances on Various I2I Tasks**: Through extensive experimentation on various I2I tasks and image resolutions, we show that DBMSolver consistently achieves state-of-the-art results. It surpasses the current state-of-the-art works (Zheng et al., 2024; Li et al., 2023) in terms of image quality and computational efficiency.

The source code will be made publicly available for transparency and to further support new research.

## 2 PRELIMINARIES AND RELATED WORK

### 2.1 DIFFUSION-BASED GENERATIVE MODELS

**Diffusion Probabilistic Models (DPMs).** Owing to their ability to generate high-quality outputs, DPMs have become ubiquitous for various noise-to-image generation tasks (Rombach et al., 2022; Karras et al., 2022). DPMs learn to traverse from a Gaussian distribution $p_{\text{prior}}(\mathbf{x})$ to an unknown data distribution $p_0(\mathbf{x}) := p_{\text{data}}(\mathbf{x})$ through a gradual *denoising process* (Ho et al., 2020; Song et al., 2020b; Dhariwal & Nichol, 2021). In other words, starting from a prior distribution $p_T(\mathbf{x}) := p_{\text{prior}}(\mathbf{x}) \approx \mathcal{N}(\mathbf{0}, \sigma_T^2 \boldsymbol{I})$ with $\sigma_T > 0$, DPMs iteratively denoise $\mathbf{x}_T \sim p_T(\mathbf{x})$ (*i.e.*, white noise) to recover the desired output $\mathbf{x}_0 \sim p_0(\mathbf{x})$. This *reverse diffusion process* is shown to follow the Ordinary Differential Equation (ODE) (Anderson, 1982; Song et al., 2020b):

$$\mathrm{d}\mathbf{x}_t = \left[ \boldsymbol{f}(\mathbf{x}_t, t) - \frac{1}{2} g(t)^2 \nabla_{\mathbf{x}_t} \log p_t(\mathbf{x}) \right] \mathrm{d}t, \tag{1}$$

where $p_t(\mathbf{x})$ is the marginal distribution of $\mathbf{x}_t$ at $t$, and $\nabla_{\mathbf{x}_t} \log p_t(\mathbf{x})$ is its *score function* learned by a neural network (Hyvärinen, 2005), and $\boldsymbol{f}(\mathbf{x}_t, t)$ and $g(t)$ are the drift and diffusion coefficients, respectively (see Section A). Song et al. (2020b) term this the *Probability Flow (PF) ODE*.

**Diffusion Bridge Models (DBMs).** Although DPMs have gained popularity for N2I Generation tasks, their underlying theory only holds when the prior distribution is purely Gaussian, *i.e.*, $p_T(\mathbf{x}) \approx \mathcal{N}(\mathbf{0}, \sigma_T^2 \boldsymbol{I})$. However, this assumption does not hold for I2I translation tasks, where $p_T(\mathbf{x})$ is not necessarily Gaussian noise, thereby limiting its applicability in such settings. To solve this, Zhou et al. (2023) were able to extend the diffusion framework from N2I Generation to I2I Translation

by making use of Doob's h-transform (Doob, 1984; Rogers & Williams, 2000). By steering the forward diffusion process almost surely to a target via Doob's h-transform, they form a *diffusion bridge* between $\mathbf{x}_0 \sim p_0(\mathbf{x})$ and $\mathbf{x}_T \sim p_T(\mathbf{x})$, yielding a *conditioned* forward diffusion process.

The corresponding reverse-time process is governed by the Bridge SDE:

$$\mathrm{d}\mathbf{x}_t = \left(\boldsymbol{f}(\mathbf{x}_t, t) - g(t)^2 [\nabla_{\mathbf{x}_t} \log p_t(\mathbf{x}_t \mid \mathbf{x}_0, \mathbf{x}_T) - \nabla_{\mathbf{x}_t} \log p_t(\mathbf{x}_T \mid \mathbf{x}_t)]\right) \mathrm{d}t + g(t)\,\mathrm{d}\mathbf{w}_t, \quad (2)$$

where $\nabla_{\mathbf{x}_t} \log p_t(\mathbf{x}_t \mid \mathbf{x}_0, \mathbf{x}_T)$ is the score of the tractable *conditional probability*, $p_t(\mathbf{x}_t \mid \mathbf{x}_0, \mathbf{x}_T)$:

$$\nabla_{\mathbf{x}_t} \log p_t(\mathbf{x}_t \mid \mathbf{x}_0, \mathbf{x}_T) = \frac{\frac{\mathrm{SNR}_T}{\mathrm{SNR}_t}\frac{\alpha_t}{\alpha_T}\mathbf{x}_T + \alpha_t\left(1 - \frac{\mathrm{SNR}_T}{\mathrm{SNR}_t}\right)\mathbf{x}_0 - \mathbf{x}_t}{\sigma_t^2\left(1 - \frac{\mathrm{SNR}_T}{\mathrm{SNR}_t}\right)}. \quad (3)$$

This score is learned by a DBM via Bridge Score Matching (Zhou et al., 2023) (*i.e.*, $\boldsymbol{s_\theta}(\mathbf{x}_t, t, \mathbf{x}_T) \approx \nabla_{\mathbf{x}_t} \log p_t(\mathbf{x}_t \mid \mathbf{x}_0, \mathbf{x}_T)$). The score of the *transition probability*, $p_t(\mathbf{x}_T \mid \mathbf{x}_t)$, is given by:

$$\nabla_{\mathbf{x}_t} \log p_t(\mathbf{x}_T \mid \mathbf{x}_t) = \frac{\frac{\alpha_t}{\alpha_T}\mathbf{x}_T - \mathbf{x}_t}{\sigma_t^2\left(\frac{\mathrm{SNR}_t}{\mathrm{SNR}_T} - 1\right)}, \qquad \mathrm{SNR}_t := \alpha_t^2/\sigma_t^2, \quad (4)$$

where $\mathrm{SNR}_t$ is the signal-to-noise ratio at time $t$. Lastly, the SDE in Equation (2) has an equivalent ODE interpretation, which we name *"Bridge Probability Flow (PF) ODE"*:

$$\mathrm{d}\mathbf{x}_t = \left[\boldsymbol{f}(\mathbf{x}_t, t) - g(t)^2\left(\frac{1}{2}\nabla_{\mathbf{x}_t} \log p_t(\mathbf{x}_t \mid \mathbf{x}_0, \mathbf{x}_T) - \nabla_{\mathbf{x}_t} \log p_t(\mathbf{x}_T \mid \mathbf{x}_t)\right)\right] \mathrm{d}t. \quad (5)$$

### 2.2 FAST SAMPLERS FOR DIFFUSION-BASED MODELS

For DPM-based N2I Generation, works such as Lu et al. (2022a;b); Zhao et al. (2024) proposed fast samplers that generate high-quality images in $\leq 20$ NFEs. These methods follow the assumption that the prior is a pure Gaussian distribution. However, since this assumption becomes **invalid** for I2I Translation (as prior $p_T(\mathbf{x})$ can be arbitrary), their theoretical foundation is unsuitable for I2I tasks, calling for samplers that support arbitrary priors. We summarize this in Table 1.

Meanwhile, to generate high-fidelity images with DBMs, Zhou et al. proposed the Hybrid Heun (HH) Sampler– which alternatively solves the Bridge SDE (Equation (2)) via the 1st-order Euler-Maruyama method, and the Bridge PF ODE (Equation (5)) via the 2nd-order Heun method. Next, inspired by Song et al. (2020a), Zheng et al. (2024) recently proposed a non-Markovian 1st-order solver called DBIM. Although DBIM improves the sampling speed, it still requires dozens of NFEs for high-quality images. In contrast, we analyze and rigorously derive exact solutions to the Bridge SDE and PF ODE to propose a **higher-order** sampler that surpasses DBIM in image quality and efficiency.

Table 1: DPMs assume $p_T(\mathbf{x}) \approx \mathcal{N}(\mathbf{0}, \sigma_T^2\mathbf{I})$, preventing arbitrary $p_T(\mathbf{x})$ thus unsuitable for DBMs.

| Sampling Method | Prior Sample $\mathbf{x}_T$ | Theoretically valid on Image-to-Image Translation | Sampling Procedure | Is Markovian |
|---|---|---|---|---|
| *Samplers designed for N2I-based DPMs:* | | | | |
| DDIM [33] | $\mathbf{x}_T \sim \mathcal{N}(\mathbf{0}, \sigma_T^2\boldsymbol{I})$ | ✗ | $p_t(\mathbf{x}_{t_{i-1}} \mid \mathbf{x}_{t_i})$ | ✗ |
| DPMSolver++2M [26] | $\mathbf{x}_T \sim \mathcal{N}(\mathbf{0}, \sigma_T^2\boldsymbol{I})$ | ✗ | Exact Soln. of ODE via [11] | ✓ |
| *Samplers designed for I2I-based DBMs:* | | | | |
| Hybrid Heun [45] | $\mathbf{x}_T \sim p_{\mathrm{prior}}(\mathbf{x})$ | ✓ | SDE (Euler-Maruyama) & ODE (Heun) | ✓ |
| DBIM [44] | $\mathbf{x}_T \sim p_{\mathrm{prior}}(\mathbf{x})$ | ✓ | $p_t(\mathbf{x}_{t_{i-1}} \mid \mathbf{x}_{t_i}, \mathbf{x}_T)$ | ✗ |
| DBMSolver **(Ours)** | $\mathbf{x}_T \sim p_{\mathrm{prior}}(\mathbf{x})$ | ✓ | Exact Soln. of Bridge SDE & ODE via [11] | ✓ |

## 3 DBMSOLVER: DEVISING A FAST SAMPLER FOR DBMS

The fundamental difference between DPMs and DBMs is the fact that $\mathbf{x}_T$ is pure noise for DPMs, but it can be an arbitrary image for DBMs. Consequently, DBMs involve a reverse diffusion process conditioned on $\mathbf{x}_T$, which is crucial for I2I Translation. As discussed in Section 2.2, this crucial distinction invalidates the direct application of state-of-the-art fast N2I solvers (such as DPMSolver++ (Lu et al., 2022b)) for sampling DBMs. We explore a different approach to develop fast

samplers specifically for DBMs by thoroughly analyzing their underlying reverse diffusion SDE and ODE (Equations 2 & 5) and deriving their exact solutions using Exponential Integrators (Hochbruck & Ostermann, 2010). With these solutions, we develop a higher-order sampling procedure that generates high-quality images significantly faster, tailor-made for DBMs. Given prior and target images $\mathbf{x}_T$ and $\mathbf{x}_0$, let $\boldsymbol{D}_{\boldsymbol{\theta}}(\mathbf{x}_s, s, \mathbf{x}_T, T)$ denote an $\mathbf{x}$-predicting DBM such that $\boldsymbol{D}_{\boldsymbol{\theta}}(\mathbf{x}_s, s, \mathbf{x}_T, T) \approx \mathbf{x}_0$ for $s \in [0, T]$. For brevity, we write $\boldsymbol{D}_{\boldsymbol{\theta}}(\mathbf{x}_s) := \boldsymbol{D}_{\boldsymbol{\theta}}(\mathbf{x}_s, s, \mathbf{x}_T, T)$.

### 3.1 Deriving the Solution to the Bridge SDE

We analyze and derive an exact solution to the Bridge SDE (Equation (2)) by leveraging the Exponential Integrators (EI) method (Hochbruck & Ostermann, 2010), which is particularly powerful for *semi-linear* differential equations of the form $\frac{d\mathbf{x}_t}{dt} = L(t)\,\mathbf{x}_t + N(\mathbf{x}_t, t)$, where $L(t)$ and $N(\mathbf{x}_t, t)$ are linear and non-linear coefficients, respectively. Simplifying and re-structuring the SDE, we observe that it indeed has a semi-linear structure, allowing us to utilize the EI method to obtain an exact $1^{\text{st}}$-order solution. The Bridge SDE fits this form (proof in Section B.1), allowing for an accurate sampling procedure, as described in Proposition 1.

**Proposition 1** *Given an initial value $\mathbf{x}_s$ and time steps $0 \leq t < s \leq T$, the exact solution to $\mathbf{x}_t$ is:*

$$\mathbf{x}_t = \frac{SNR_s}{SNR_t}\frac{\alpha_t}{\alpha_s}\mathbf{x}_s + \alpha_t\left(1 - \frac{SNR_s}{SNR_t}\right)\mathbf{D}_{\boldsymbol{\theta}}(\mathbf{x}_s) + \sigma_t\sqrt{1 - \frac{SNR_s}{SNR_t}}\,\mathbf{z}_t, \tag{6}$$

*where $\mathbf{z}_t \sim \mathcal{N}(\mathbf{0}, \mathbf{I})$, and $SNR_t := \alpha_t^2/\sigma_t^2$ is the signal-to-noise ratio at time $t$.*

### 3.2 Deriving the Solution to the Bridge PF ODE

Having set grounds with Proposition 1, we next focus on the Bridge PF ODE (Equation (5)). Similar to the Bridge SDE analysis above, we show that the Bridge ODE also exhibits semi-linearity in its structure, which has been largely overlooked in prior works. We take advantage of this semi-linearity by deriving a closed-form exact solution through the EI method. Then, we utilize the change-of-variables method to reformulate the solution as an exponentially-weighted integral. Finally, we analytically minimize the discretization errors by Taylor expanding this integral, yielding a fast and efficient sampling procedure, as presented in Proposition 2 with its proof in Section B.2.

**Proposition 2** *Given an initial value $\mathbf{x}_s$ and time steps $0 \leq t < s < T$, the exact solution to $\mathbf{x}_t$ is:*

$$\mathbf{x}_t = \frac{\alpha_t}{\alpha_s}e^{2(\lambda_s - \lambda_t)}\sqrt{\frac{\rho(\lambda_t, \lambda_T)}{\rho(\lambda_s, \lambda_T)}}\,\mathbf{x}_s + \frac{\alpha_t}{\alpha_T}e^{2(\lambda_T - \lambda_t)}\left[1 - \sqrt{\frac{\rho(\lambda_t, \lambda_T)}{\rho(\lambda_s, \lambda_T)}}\right]\mathbf{x}_T$$

$$+ \alpha_t\,e^{-2\lambda_t}\sqrt{\rho(\lambda_t, \lambda_T)}\underbrace{\int_{\lambda_s}^{\lambda_t}\frac{e^{2\lambda}\,\mathbf{D}_{\boldsymbol{\theta}}(\mathbf{x}_\lambda)}{\sqrt{\rho(\lambda, \lambda_T)}}\,d\lambda}_{\textit{The Exponential Integral}}, \tag{7}$$

*where $\lambda_t := \log(\alpha_t/\sigma_t)$ with the inverse function $t_\lambda(\cdot)$, and $\mathbf{x}_\lambda := \mathbf{x}_{t_\lambda(\lambda)}$ is the change-of-variable form for $\lambda$, and $\rho(a, b) := e^{2(a-b)} - 1$. Intuitively, $\lambda_t$ can be thought of as half the log SNR at time $t$.*

We simplify the *Exponential Integral* in Equation (7) by taking its $(k-1)^{\text{th}}$ Taylor expansion:

$$\int_{\lambda_a}^{\lambda_b}\frac{e^{2\lambda}\,\boldsymbol{D}_{\boldsymbol{\theta}}(\mathbf{x}_\lambda)}{\sqrt{\rho(\lambda, \lambda_T)}}\,d\lambda \approx \sum_{n=0}^{k-1}\underbrace{\boldsymbol{D}_{\boldsymbol{\theta}}^{(n)}(\mathbf{x}_{\lambda_s})}_{\text{Estimated}}\underbrace{\int_{\lambda_s}^{\lambda_t}\frac{e^{2\lambda}}{\sqrt{\rho(\lambda, \lambda_T)}}\frac{(\lambda - \lambda_s)^n}{n!}d\lambda}_{\text{Analytically Computed (Section B.3)}} + \underbrace{\mathcal{O}((\lambda_t - \lambda_s)^{k+1})}_{\text{Omitted}}, \tag{8}$$

where $k \geq 1$, and $\boldsymbol{D}_{\boldsymbol{\theta}}^{(n)}(\mathbf{x}_{\lambda_s}) := \frac{d^n \boldsymbol{D}_{\boldsymbol{\theta}}(\mathbf{x}_{\lambda_s})}{d\lambda^n}$ is the $n^{\text{th}}$-order derivative of $\boldsymbol{D}_{\boldsymbol{\theta}}(\cdot)$ w.r.t. $\lambda$. Note that we omit the error term $\mathcal{O}((\lambda_t - \lambda_s)^{k+1})$. In Section B.4, we demonstrate that our solver generalizes DBIM (Zheng et al., 2024), and both methods converge when $k = 1$.

### 3.3 DEVISING DBMSOLVER USING EQUATIONS 6 AND 7

**Initial Step.** In the sampling procedure, the initial step is taken from time $s = T$ to time $t = T - \epsilon$, where $\epsilon$ is a small value. However, the exact solution of the Bridge PF ODE in Equation (7) is only valid for $s < T$, which implies that it cannot be employed for this initial step (*i.e.*, from $s = T$ to $t = T - \epsilon$). This is because if we were to implement Equation (7) for the initial step, then $\rho(\lambda_s, \lambda_T) = \rho(\lambda_T, \lambda_T) = 0$, which would cause the coefficient of $\mathbf{x}_s$ to diverge to infinity, thereby exhibiting a singularity. Thus, we instead employ the exact solution of the Bridge SDE (Equation (6)) exclusively for the initial step, while applying the exact solution of the Bridge PF ODE (Equation (7)) for the subsequent steps (where $s \leq T - \epsilon$ and $t < s$), as described next.

**Subsequent Steps.** Higher-order formulations of Equation (8) can lead to a sampling procedure capable of generating high-quality images more efficiently, as demonstrated in previous works (Lu et al., 2022a;b; Zhao et al., 2024). This improvement stems from the fact that higher-order Taylor expansions have reduced error bounds, yielding more accurate approximations. Following this idea, we set $k = 2$, resulting in an exact 2nd-order Bridge PF ODE solution for Equation (7). We adopt this 2nd-order formulation for DBMSolver and describe the complete derivation in Section B.5.

**Summarizing the Algorithm.** Given time steps $T = t_N > t_{N-1} > \cdots > t_1 > t_0 = 0$, we first compute $\tilde{\mathbf{x}}_{t_{N-1}}$ from prior image $\mathbf{x}_{t_N} \sim p_T(\mathbf{x})$ using Equation (6). For the next $N - 2$ steps, we iteratively apply Equation (7) with $k = 2$, yielding better approximations for each intermediate noisy sample until $\tilde{\mathbf{x}}_{t_1}$. To obtain the final $\tilde{\mathbf{x}}_0$ prediction, we solve the Bridge PF ODE from $t_1$ to $t_0$ using the widely used Euler method, resulting in a high-fidelity output. We summarize it in Algorithm 1 and validate it empirically in the next section.

---

**Algorithm 1** DBMSolver: A Fast Sampler for Diffusion-based I2I Translation

---

**Inputs:** Pretrained DBM $\boldsymbol{D_\theta}(\cdot)$, Number of sampling steps $N$, Time steps $T = t_N > \cdots > t_1 > t_0 = 0$, and Prior distribution $p_T(\mathbf{x})$.

**Initialization:** Sample $\tilde{\mathbf{x}}_T \sim p_T(\mathbf{x})$, $\mathbf{z} \sim \mathcal{N}(\mathbf{0}, \boldsymbol{I})$, and $\tilde{\mathbf{x}}_0 \leftarrow \boldsymbol{D_\theta}(\tilde{\mathbf{x}}_{t_N})$

**Initial Stochastic Update:** Calculate $\tilde{\mathbf{x}}_{t_{N-1}}$ from $\tilde{\mathbf{x}}_T$ using Equation (6).

**Subsequent Deterministic Refinement:**
**for** $i = N - 1$ **to** $1$ **do**
  **if** $i > 1$ **then**
    $a \leftarrow t_i$, and $b \leftarrow t_{i-1}$.
    Calculate $\tilde{\mathbf{x}}_b$ from $\tilde{\mathbf{x}}_a$ using Equation (7) (with $k = 2$).  {▷ Refer Section B.5.}
  **else**
    $\mathrm{d}\mathbf{x}_{t_1} \leftarrow \boldsymbol{f}(\tilde{\mathbf{x}}_{t_1}, t_1) - g(t_1)^2 \left( \frac{1}{2} \boldsymbol{D_\theta}(\tilde{\mathbf{x}}_{t_1}) - \nabla_{\mathbf{x}_t} \log p_{t_1}(\mathbf{x}_T \mid \tilde{\mathbf{x}}_{t_1}) \right)$
    $\tilde{\mathbf{x}}_0 \leftarrow \tilde{\mathbf{x}}_{t_1} + (t_0 - t_1) \, \mathrm{d}\mathbf{x}_{t_1}$  {▷ Final Euler Update}
  **end if**
**end for**
**Output:** $\tilde{\mathbf{x}}_0$  {▷ Final translated image}

---

## 4 EXPERIMENTS AND RESULTS

We conducted extensive experiments to evaluate DBMSolver against established baselines on various I2I Translation tasks, including conditional image inpainting and semantics-to-image generation, to demonstrate its versatility across diverse tasks. Specifically, we evaluated on the following challenging datasets: Sketch-to-Image on *Edges2Handbags* (E2H) (Isola et al., 2017), Surface normals-to-Image on *DIODE-Outdoor* (Vasiljevic et al., 2019), Face-to-Comic stylization on *Face2Comics* (F2C), Conditional Image Inpainting with central masks on *ImageNet* (Deng et al., 2009), and Semantic Label-to-Face generation on *CelebAMask-HQ* (Lee et al., 2020).

We mainly assess sampling quality using FID (Heusel et al., 2017), and computational efficiency via the number of forward evaluations NFEs (Song et al., 2020a; Lu et al., 2022a). For CelebAMask-HQ, we additionally report classification accuracy (CA), following Li et al. (2023). We use the publicly available DBM checkpoints from Zhou et al. (2023) for E2H and DIODE, highlighting DBMSolver's

Table 2: Quantitative results on DIODE and Edges2Handbags. FID (↓) is reported against NFE (↓). Time denotes total sampling duration in minutes; Rate indicates images generated/second.

| Family | Method | NFE (↓) | DIODE (256×256) [40] | | | Edges2Handbags (64×64) [14] | | |
|---|---|---|---|---|---|---|---|---|
| | | | Time (↓) | Rate (↑) | FID (↓) | Time (↓) | Rate (↑) | FID (↓) |
| GAN | Pix2Pix [14] | 1 | – | – | 82.40 | – | – | 74.80 |
| Diffusion & Flow | DDIB [36] | ≥ 40 | – | – | 242.30 | – | – | 186.84 |
| | SDEdit [27] | ≥ 40 | – | – | 31.14 | – | – | 26.50 |
| | Rectified Flow [23] | ≥ 40 | – | – | 25.30 | – | – | 77.18 |
| | I²SB [22] | ≥ 40 | – | – | 9.34 | – | – | 7.43 |
| Sampling via Diffusion Bridge Models | DPMSolver++2M† [26] | 100 | 237.68 | 1.15 | 98.68 | 199.16 | 11.59 | 33.33 |
| | Hybrid Heun [45] | 119 | 283.54 | 0.97 | 4.43 | 327.58 | 7.05 | 1.83 |
| | DBIM [44] | 6 | 19.21 | 14.31 | 12.13 | 12.05 | 191.53 | 3.27 |
| | | 20 | 47.63 | 5.77 | 4.99 | 39.98 | 57.76 | 1.74 |
| | | 100 | 238.32 | 1.15 | 2.57 | 199.63 | 11.57 | 0.89 |
| | DBMSolver (Ours) | 6 | 14.18 | 19.38 | 3.38 | 8.33 | 276.99 | 0.97 |
| | | 10 | 23.48 | 11.71 | _2.15_ | 13.85 | 166.67 | 0.58 |
| | | 20 | 46.85 | 5.87 | **2.06** | 27.64 | 83.53 | _0.54_ |
| | | 30 | 70.14 | 3.92 | **2.06** | 41.43 | 55.73 | **0.52** |

† denotes sampler specifically designed for N2I Generation

training-free integration. For ImageNet inpainting, we adopt the DBM checkpoint from Zheng et al. (2024), which was finetuned via I²SB (Liu et al., 2023) from a pre-trained N2I Diffusion Model. For datasets lacking checkpoints (e.g., Face2Comics, CelebAMask-HQ), we train DBMs from scratch using the ADM U-Net (Dhariwal & Nichol, 2021), following standard diffusion architectures.

Our implementation builds on the official DBIM codebase (Zheng et al., 2024); training details are in Section C.1. We benchmark DBMSolver against current state-of-the-art I2I translation methods. Our main baseline is DDBM (Hybrid Heun) (Zhou et al., 2023), using reported metrics for fair comparison with prior GANs and diffusion models. We also include DBIM (Zheng et al., 2024), a non-Markovian DBM accelerator, and N2I-Generation-based DPM-Solver++ (Lu et al., 2022b). Additional baselines of DDIB (Su et al., 2022), SDEdit (Meng et al., 2021), Rectified Flow (Liu et al., 2022), and I²SB (Liu et al., 2023) are evaluated following the DDBM and DBIM protocols.

### 4.1 RESULTS

**Image Translation on E2H (64×64) and DIODE (256×256).** Table 2 reports FID scores and NFEs across methods. DBMSolver achieves state-of-the-art results with significantly fewer evaluations. At just 10 NFEs, it achieves FID scores of 0.58 (E2H) and 2.15 (DIODE), outperforming both Hybrid Heun and DBIM. Its high efficiency at low NFEs enables rapid sampling, making it well-suited for real-time DBM applications by supporting faster generation and higher throughput. It exhibits strong scalability with increasing NFEs, yielding further improvements in FID.

The trends in Figure 2-a,b show that as NFE increases, DBMSolver quickly achieves high fidelity and remains stable. Figure 3 supports this, indicating that even at low NFEs (e.g., 6), DBMSolver and DBIM generate visually rich, coherent outputs, outperforming others in detail and realism. DPMSolver++2M preserves structure but lacks vibrant color and texture, especially at lower NFEs. While DBIM yields appealing outputs, it lacks fine detail compared to our method– a gap reflected in FID and trend metrics. Please refer to the intricate structural details observable in the tree branches and twigs within the DIODE images, as well as the fine-grained textures and contours present in the handbag depictions. DBMSolver consistently balances efficiency and quality across datasets.

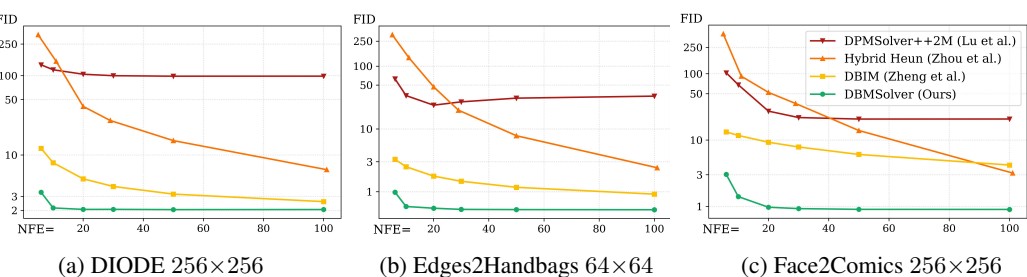

(a) DIODE 256×256        (b) Edges2Handbags 64×64        (c) Face2Comics 256×256

Figure 2: FID vs. NFE on different datasets. We consistently get better FID scores with fewer NFEs.

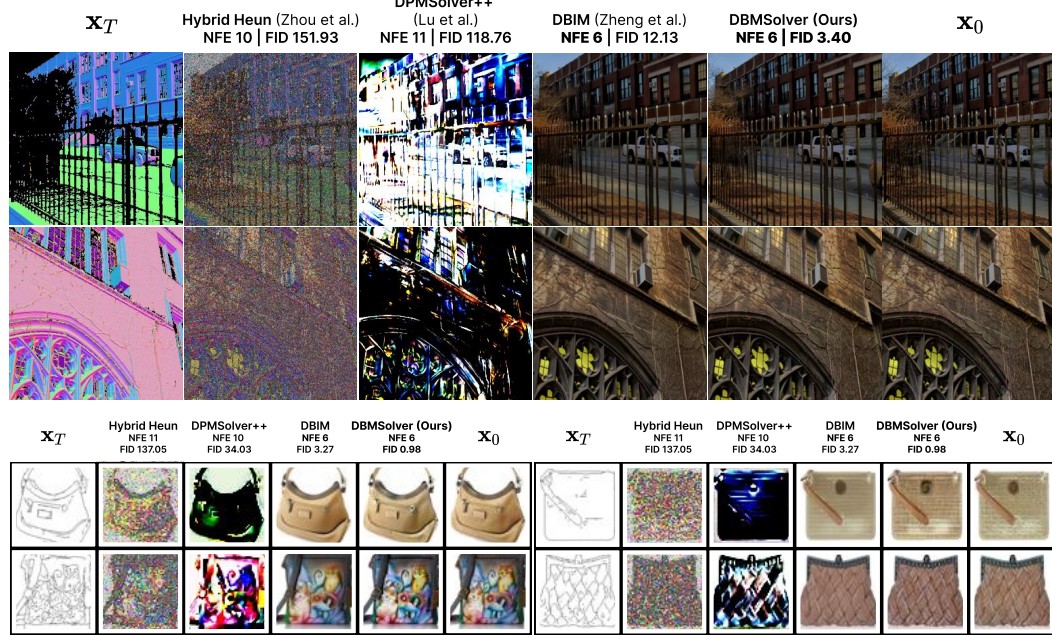

Figure 3: Visuals for Table 2; DPMSolver++ and HH shown at 11 NFEs due to poor 6-NFE quality.

**Label-to-Face Generation on CelebAMask-HQ ($256 \times 256$).** Figures 4 and 5 and Table 4 show that our method generates images with precise facial segmentation and coherent boundaries. At as low as 6 NFEs, DBMSolver achieves a FID of 34.76 outperforming DBIM's 44.92 as well as GAN-based models and other diffusion approaches, while using significantly fewer NFEs. Visually, DBMSolver preserves fine structural details such as eye contours, hairlines, and mask edges, which are often blurred or distorted in DBIM outputs. DBMSolver consistently produces sharper, more anatomically faithful generations, enhancing both realism and image accuracy.

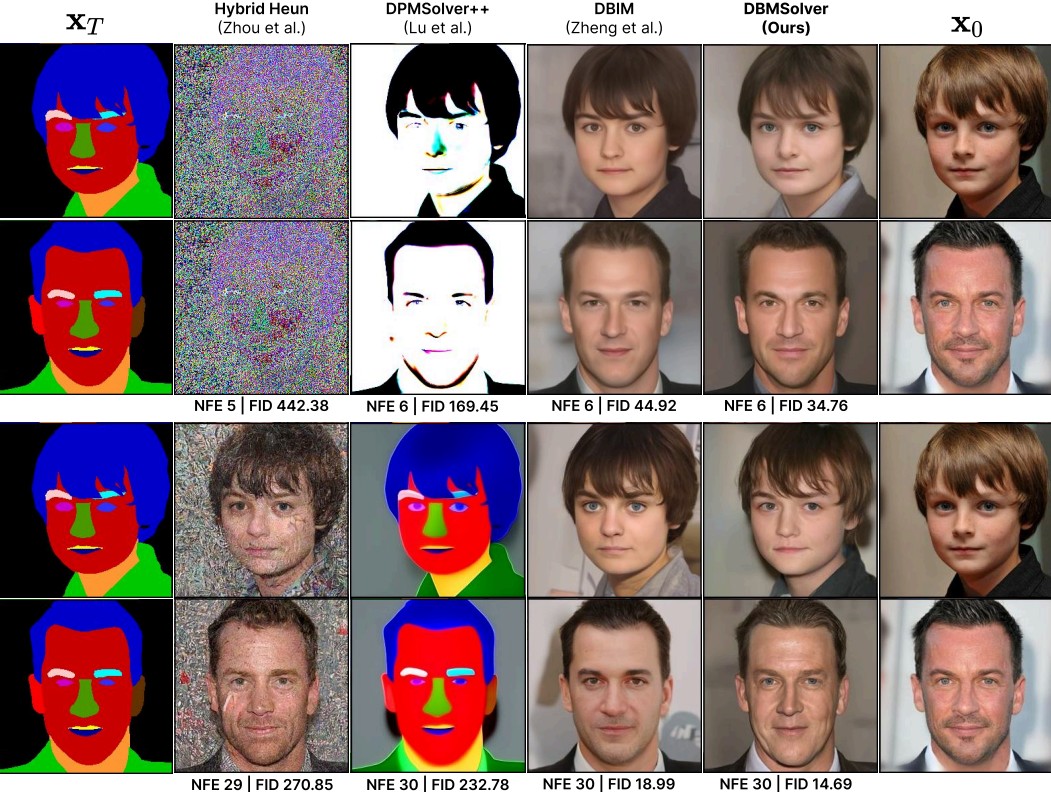

Figure 4: Label-to-Face Generation on CelebAMask-HQ $256 \times 256$.

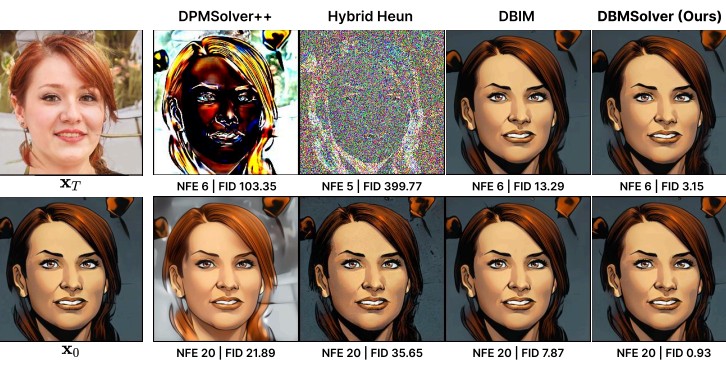

Figure 5: Generated samples on CelebAMask-HQ $256 \times 256$ using our DBMSolver in 6 NFEs.

Figure 6: Image Stylization on Face2Comics $256 \times 256$.

Table 3: Quantitative comparison on Face2Comics.

| Method | NFE ($\downarrow$) | FID ($\downarrow$) |
|---|---|---|
| *GANs & Other Diffusion-based Models:* | | |
| Pix2Pix [14] | 1 | 49.96 |
| CycleGAN [47] | 1 | 35.13 |
| DRIT++ [20] | – | 28.87 |
| CDE [32] | – | 33.98 |
| LDM [30] | – | 24.28 |
| BBDM [21] | 200 | 23.20 |
| *Sampling via Diffusion Bridge Models:* | | |
| DPMSolver++2M[†] [26] | 20 | 27.34 |
| | 30 | 21.88 |
| Hybrid Heun [45] | 119 | 2.36 |
| DBIM [44] | 6 | 13.29 |
| | 10 | 11.75 |
| | 20 | 9.28 |
| | 30 | 7.87 |
| DBMSolver (Ours) | 6 | 3.04 |
| | 10 | 1.40 |
| | 20 | 0.97 |
| | 30 | **0.92** |

Table 4: Quantitative results for Label-to-Face Generation on CelebAMask-HQ at NFEs of 6 and 30, complementing the visual examples in Fig. 4.

| Methods | CelebAMask-HQ ($256 \times 256$) [19] | | | |
|---|---|---|---|---|
| | Time | Rate | NFE ($\downarrow$) | FID ($\downarrow$) |
| *GANs & Other Diffusion-based Models:* | | | | |
| Pix2Pix [14] | – | – | 1 | 56.99 |
| CycleGAN [47] | – | – | 1 | 78.23 |
| DRIT++ [20] | – | – | | 77.79 |
| SPADE [28] | – | – | | 44.17 |
| OASIS [38] | – | – | | 27.75 |
| CDE [32] | – | – | | 24.40 |
| LDM [30] | – | – | | 22.81 |
| BBDM [21] | – | – | 200 | 21.35 |
| *Sampling via Diffusion Bridge Models:* | | | | |
| DPMSolver++2M[†] [26] | 68.65 | 5.87 | 20 | 223.75 |
| | 103.14 | 3.90 | 30 | 232.78 |
| Hybrid Heun [45] | 409.84 | 0.98 | 119 | 97.75 |
| DBIM [44] | 20.64 | 19.52 | 6 | 44.92 |
| | 34.51 | 11.67 | 10 | 34.18 |
| | 68.84 | 5.85 | 20 | 23.30 |
| | 104.24 | 3.86 | 30 | 18.99 |
| DBMSolver (Ours) | 20.44 | 19.71 | 6 | 34.76 |
| | 33.90 | 11.88 | 10 | 24.93 |
| | 67.67 | 5.95 | 20 | 17.68 |
| | 102.65 | 3.93 | 30 | **14.69** |

Table 5: Quantitative results for Class-Conditional Inpainting (center $128 \times 128$ mask) on ImageNet. DBMSolver achieves superior FID and Classification Accuracy (CA) across all NFEs, delivering high image fidelity with only 6 NFEs, outperforming prior methods that require more NFEs for comparable quality.

| Methods | ImageNet ($256 \times 256$) [2] | | | | |
|---|---|---|---|---|---|
| | Time | Rate | NFE ($\downarrow$) | FID ($\downarrow$) | CA ($\uparrow$) |
| *Other Diffusion-based Models:* | | | | | |
| DDRM [18] | – | – | 20 | 24.40 | 62.1 |
| ΠGDM [34] | – | – | 100 | 7.30 | 72.6 |
| DDNM [41] | – | – | 100 | 15.10 | 55.9 |
| Palette [31] | – | – | 1000 | 6.10 | 63.0 |
| I²SB [22] | – | – | 1000 | 4.90 | 66.1 |
| *Sampling via Diffusion Bridge Models:* | | | | | |
| DPMSolver++2M[†] [26] | 29.38 | 5.67 | 20 | 37.99 | 51.9 |
| | 43.91 | 3.79 | 30 | 36.68 | 52.3 |
| Hybrid Heun [45] | 172.78 | 0.96 | 119 | 6.02 | 69.5 |
| DBIM [44] | 8.84 | 18.83 | 6 | 5.36 | 70.2 |
| | 14.69 | 11.33 | 10 | 4.50 | 71.8 |
| | 29.39 | 5.66 | 20 | 4.13 | 71.9 |
| | 44.11 | 3.77 | 30 | 4.04 | 71.9 |
| DBMSolver (Ours) | 8.75 | 18.88 | 6 | 5.02 | 70.7 |
| | 14.60 | 11.33 | 10 | 4.38 | 71.2 |
| | 29.18 | 5.66 | 20 | 4.07 | 72.0 |
| | 43.76 | 3.80 | 30 | **4.03** | **72.4** |

**Image Stylization on Face2Comics ($256 \times 256$).** As shown in Table 3 and Figure 6, DBMSolver achieves top performance with just 10 NFEs. At 30 NFEs, it attains an FID of 0.92, outperforming Hybrid Heun (2.36 at 119 NFEs), DBIM (7.87 at 30 NFEs), and various GAN and diffusion methods. These results highlight DBMSolver's efficiency and sample quality across diverse datasets, as further illustrated in Figure 1. Even at 6 NFEs, its outputs rival higher-NFE baselines, demonstrating strong perceptual fidelity at minimal cost.

**Class-Conditional Inpainting on ImageNet ($256 \times 256$).** Table 5 and Fig. 7 demonstrate DBMSolver's superior performance. At just 6 NFEs, it achieves strong semantic coherence and texture synthesis, surpassing methods requiring hundreds of steps. At 30 NFEs, it attains the best FID (4.03) and top classification accuracy (72.4%), highlighting its efficiency and quality. DBMSolver

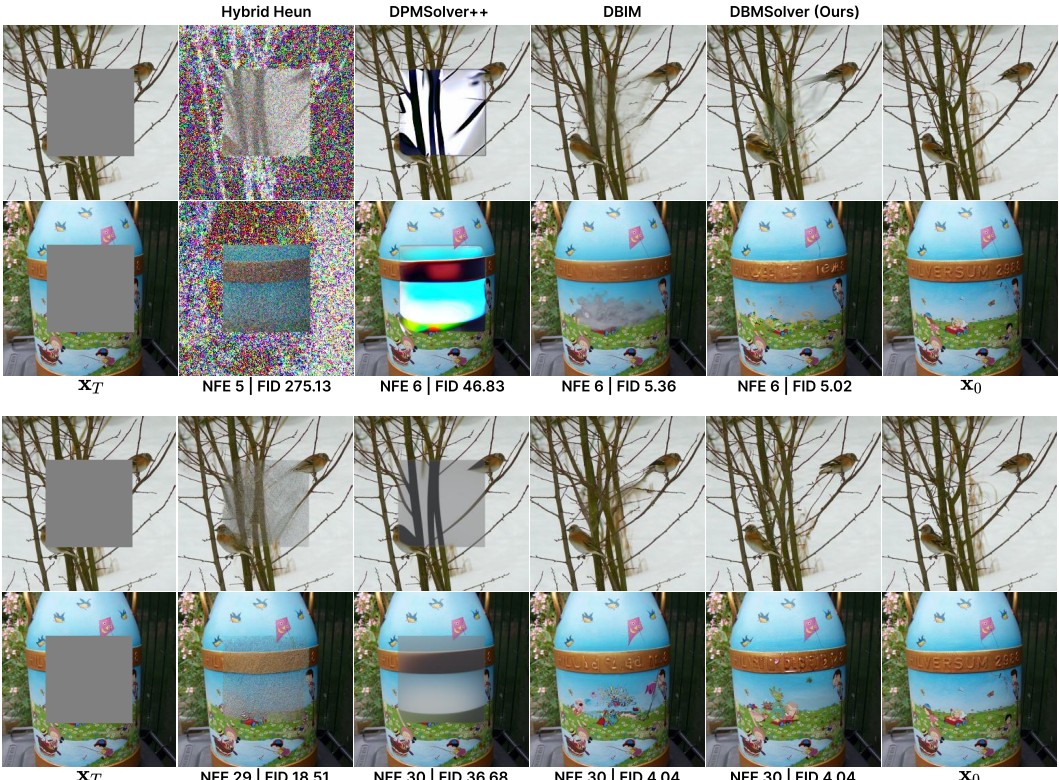

Figure 7: Class-Conditional Inpainting on ImageNet $256 \times 256$.

also avoids the blurry textures seen in DBIM—evident in the sparrow's feathers and branches—and maintains structural fidelity in cases like the milk barrel, where DBIM hallucinates unrealistic content.

**Efficiency Analysis via Time and Sampling Rate.** Beyond superior FID and classification accuracy, DBMSolver demonstrates exceptional efficiency in runtime and throughput. As shown in Table 4 and Table 5, DBMSolver achieves high-quality results with minimal computational cost. At 30 NFEs, it completes sampling in just 102.65s on CelebAMask-HQ and 43.76s on ImageNet, significantly faster than Hybrid Heun (409.84s and 172.78s, respectively). Moreover, DBMSolver maintains a high sampling rate across all NFE budgets, peaking at 19.71 samples/min for 6 NFEs on CelebAMask-HQ and 11.33 samples/min for 10 NFEs on ImageNet, nearly double that of DBIM and vastly exceeding other bridge-based samplers. These metrics highlight DBMSolver's ability to balance speed and quality, i.e., real-time generation without compromising visual fidelity.

**Limitations and Future Work.** Our discrete-time DBMs may still face discretization errors. Continuous-time diffusion models (Karras et al., 2022; 2024; Sun et al., 2022) may help address this. Further research could investigate exponential Rosenbrock-type methods (Hochbruck et al., 2009) for DBMs to improve generation quality and efficiency. Exploring DBMs for more complex tasks, like text-conditioned I2I translation, is also a promising avenue for research.

## 5 CONCLUSION

In conclusion, we introduce DBMSolver, a principled, training-free method that significantly enhances the efficiency and quality of diffusion-based I2I translation. By leveraging the semi-linear structure of the Bridge SDE and PF ODE, DBMSolver accelerates sampling without compromising fidelity. Experiments on diverse datasets, such as Edges2Handbags, DIODE-Outdoor, Face2Comics, CelebAMask-HQ, and Conditional ImageNet Inpainting, show that DBMSolver achieves high-quality results with far fewer NFEs, setting a new benchmark for efficient diffusion bridge models. This work marks a step toward the practical deployment of powerful I2I and restoration tools.

**Use of Large Language Models.** LLMs were employed exclusively for editorial refinement, without influencing research design or substantive content.

**Ethics Statement.** We have read and agree to the ICLR Code of Ethics (`https://iclr.cc/public/CodeOfEthics`). Our work introduces DBMSolver, a training-free sampler for accelerating diffusion bridge models in image-to-image translation. All experiments were conducted using publicly available datasets, and no human subjects or private data were involved. We took care to avoid generating or amplifying harmful, biased, or misleading content. While generative models can pose risks in misuse or misrepresentation, our method is designed to improve computational efficiency and fidelity without introducing new ethical concerns. We encourage responsible use and transparent reporting when deploying such models in real-world applications.

**Reproducibility Statement.** We have made every effort to ensure the reproducibility of our results. Details of the DBMSolver algorithm, including its mathematical formulation and implementation, are provided in the main paper and Appendix. All datasets used are publicly available and referenced appropriately. We include a complete description of evaluation protocols and metrics. DBMSolver's pseudo-algorithm is thoroughly described in the main text and the Appendix, and we intend to publish the source code with reproduction instructions.

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

## A DIFFERENT FORMULATIONS FOR DIFFUSION MODELS

The reverse diffusion process is given by the PF ODE (Anderson, 1982; Song et al., 2020b):

$$\mathrm{d}\mathbf{x}_t = \left[ \boldsymbol{f}(\mathbf{x}_t, t) - \frac{1}{2} g(t)^2 \nabla_{\mathbf{x}_t} \log p_t(\mathbf{x}) \right] \mathrm{d}t, \tag{9}$$

where the marginal distribution of $\mathbf{x}_t$ at $t$ is $p_t(\mathbf{x})$, and $\nabla_{\mathbf{x}_t} \log p_t(\mathbf{x})$ is its *score function*, which is learned by a neural network (Hyvärinen, 2005). Furthermore, the drift and diffusion coefficients are:

$$\boldsymbol{f}(\mathbf{x}_t, t) = \mathbf{x}_t \frac{\mathrm{d}}{\mathrm{d}t} \log \alpha_t, \text{ and } g(t)^2 = -2\sigma_t^2 \frac{\mathrm{d}}{\mathrm{d}t} \log \left( \frac{\alpha_t}{\sigma_t} \right),$$

where $\alpha_t := \alpha(t)$ and $\sigma_t := \sigma(t)$, for time $t \in [0, T]$ (where $T > 0$).

Different formulations of $\alpha_t$ and $\sigma_t$ give rise to different formulations for the diffusion process. Prior works hand-design these to get the *variance-preserving* (VP) (Song et al., 2020b; Zhou et al., 2023), *variance-exploding* (VE) (Karras et al., 2022), and *TrigFlow* (Lu & Song, 2024) formulations. We contrast the design choices of such diffusion formulations in Table 6.

Table 6: Design choices for widely-used diffusion formulations.

| Formulation | $\alpha_t$ | $\sigma_t$ | $\boldsymbol{f}(\mathbf{x}_t, t)$ | $g(t)^2$ | $\mathrm{SNR}_t = \alpha_t^2/\sigma_t^2$ | Domain of $t$ |
|---|---|---|---|---|---|---|
| VP [35; 45] | $e^{-(0.5t^2 + 0.05t)}$ | $\sqrt{1 - e^{-(t^2 + 0.1t)}}$ | $-(t + 0.05)\mathbf{x}_t$ | $2t + 0.1$ | $1/(e^{(t^2+0.1t)}-1)$ | $[0.0001, 1]$ |
| VE [16] | $1$ | $t$ | $\mathbf{0}$ | $2t$ | $1/t^2$ | $[0.002, 80]$ |
| TrigFlow [24] | $\cos(t)$ | $\sin(t)$ | $-\tan(t)\,\mathbf{x}_t$ | $2\tan(t)$ | $\cot^2(t)$ | $[0, \pi/2]$ |

## B PROOFS & DERIVATIONS

### B.1 PROOF OF PROPOSITION 1

Given a well-trained DBM $\boldsymbol{D}_\theta(\cdot)$ that approximates data sample $\mathbf{x}_0$, we can simplify Equation (2) as:

$$\mathrm{d}\mathbf{x}_t = \left( \boldsymbol{f}(\mathbf{x}_t, t) - g(t)^2 [\nabla_{\mathbf{x}_t} \log p_t(\mathbf{x}_t \mid \mathbf{x}_T) - \nabla_{\mathbf{x}_t} \log p_t(\mathbf{x}_T \mid \mathbf{x}_t)] \right) \mathrm{d}t + g(t)\,\mathrm{d}\mathbf{w}_t$$

$$= \left( \mathbf{x}_t \frac{\mathrm{d}\log\alpha_t}{\mathrm{d}t} + 2\sigma_t^2 \frac{\mathrm{d}\log\left(\frac{\alpha_t}{\sigma_t}\right)}{\mathrm{d}t} [\nabla_{\mathbf{x}_t} \log p_t(\mathbf{x}_t \mid \mathbf{x}_T) - \nabla_{\mathbf{x}_t} \log p_t(\mathbf{x}_T \mid \mathbf{x}_t)] \right) \mathrm{d}t$$

$$+ \sigma_t \sqrt{-2 \frac{\mathrm{d}\log\left(\frac{\alpha_t}{\sigma_t}\right)}{\mathrm{d}t}}\,\mathrm{d}\mathbf{w}_t$$

$$= \left( \mathbf{x}_t \frac{\mathrm{d}\log\alpha_t}{\mathrm{d}t} + 2\sigma_t^2 \frac{\mathrm{d}\log\left(\frac{\alpha_t}{\sigma_t}\right)}{\mathrm{d}t} \left[ \frac{\frac{\alpha_T^2\sigma_t^2}{\sigma_T^2\alpha_t^2}\frac{\alpha_t}{\alpha_T}\mathbf{x}_T + \alpha_t\boldsymbol{D}_\theta(\mathbf{x}_t)\left(1 - \frac{\alpha_T^2\sigma_t^2}{\sigma_T^2\alpha_t^2}\frac{\alpha_t}{\alpha_T}\right) - \mathbf{x}_t}{\sigma_t^2\left(1 - \frac{\alpha_T^2\sigma_t^2}{\sigma_T^2\alpha_t^2}\frac{\alpha_t}{\alpha_T}\right)} - \frac{\frac{\alpha_t}{\alpha_T}\mathbf{x}_T - \mathbf{x}_t}{\sigma_t^2\left(\frac{\alpha_t^2\sigma_T^2}{\sigma_t^2\alpha_T^2} - 1\right)} \right] \right) \mathrm{d}t$$

$$+ \sigma_t \sqrt{-2 \frac{\mathrm{d}\log\left(\frac{\alpha_t}{\sigma_t}\right)}{\mathrm{d}t}}\,\mathrm{d}\mathbf{w}_t$$

$$= \left( \mathbf{x}_t \frac{\mathrm{d}\log\alpha_t}{\mathrm{d}t} + 2\sigma_t^2 \frac{\mathrm{d}\log\left(\frac{\alpha_t}{\sigma_t}\right)}{\mathrm{d}t} \left[ \frac{\alpha_t\boldsymbol{D}_\theta(\mathbf{x}_t) - \mathbf{x}_t}{\sigma_t^2} \right] \right) \mathrm{d}t + \sigma_t \sqrt{-2 \frac{\mathrm{d}\log\left(\frac{\alpha_t}{\sigma_t}\right)}{\mathrm{d}t}}\,\mathrm{d}\mathbf{w}_t$$

$$= \mathbf{x}_t \underbrace{\left( \frac{\mathrm{d}\log\alpha_t}{\mathrm{d}t} - 2\frac{\mathrm{d}\log\left(\frac{\alpha_t}{\sigma_t}\right)}{\mathrm{d}t} \right)}_{L(t) \text{ (Linear Term)}} \mathrm{d}t + \underbrace{\left( 2\,\alpha_t \frac{\mathrm{d}\log\left(\frac{\alpha_t}{\sigma_t}\right)}{\mathrm{d}t}\boldsymbol{D}_\theta(\mathbf{x}_t) + \sigma_t \sqrt{-2\frac{\mathrm{d}\log\left(\frac{\alpha_t}{\sigma_t}\right)}{\mathrm{d}t}}\frac{\mathrm{d}\mathbf{w}_t}{\mathrm{d}t} \right)}_{N(\mathbf{x}_t, t) \text{ (Non-linear Term)}} \mathrm{d}t,$$

$$\tag{10}$$

where $L(t)$ is the *linear* term, and $N(\mathbf{x}_t, t)$ is the *non-linear* term. This shows the semi-linearity of Equation (2). Thanks to the semi-linearity, we can make use of the Exponential Integrators method (Hochbruck & Ostermann, 2010) to solve Equation (10), as explained next.

Given initial value $\mathbf{x}_a$, where $b = a + \Delta\tau$ and $0 < b < a < T$, we obtain the solution to $\mathbf{x}_b$ as:

$$\mathbf{x}_b = e^{\int_a^b L(r)\,\mathrm{d}r}\,\mathbf{x}_a + \int_a^b e^{\int_{a+\tau}^b L(h)\,\mathrm{d}h} \cdot N(\mathbf{x}_\tau, \tau)\,\mathrm{d}\tau. \tag{11}$$

Integrating $L(r)$ with respect to $r$ from $a$ to $b$, we get:

$$\int_a^b L(r)\,\mathrm{d}r = \left[\log\frac{\sigma_r^2}{\alpha_r}\right]_a^b = \log\left(\frac{\alpha_a}{\alpha_b} \cdot \frac{\sigma_b^2}{\sigma_a^2}\right). \tag{12}$$

Next, let's define $\lambda_t := \log\frac{\alpha_t}{\sigma_t}$, with the inverse function $t_\lambda(\cdot)$, that satisfies $t_\lambda(\lambda_t) = t$. Through the *change-of-variables* method for $\lambda$, we can denote $\alpha_\lambda := \alpha_{t_\lambda(\lambda)}$, $\mathbf{x}_\lambda := \mathbf{x}_{t_\lambda(\lambda)}$, $\mathbf{D}_\theta(\mathbf{x}_\lambda) := \mathbf{D}_\theta(\mathbf{x}_{t_\lambda(\lambda)})$, $\mathbf{w}_\lambda := \mathbf{w}_{t_\lambda(\lambda)}$, $\mathrm{d}\mathbf{w}_\lambda := \sqrt{-\frac{\mathrm{d}\lambda}{\mathrm{d}t}}\,\mathrm{d}\mathbf{w}_{t_\lambda(\lambda)}$, and $N(\mathbf{x}_\lambda, \lambda) := N(\mathbf{x}_{t_\lambda(\lambda)}, t_\lambda(\lambda))$.

Thus, we re-write Equation (11) as:

$$\begin{aligned}
\mathbf{x}_b &= \frac{\alpha_a}{\alpha_b}\frac{\sigma_b^2}{\sigma_a^2}\mathbf{x}_a + \int_{\lambda_a}^{\lambda_b} \frac{\alpha_\lambda}{\alpha_b}\frac{\sigma_b^2}{\sigma_\lambda^2} \cdot N(\mathbf{x}_\lambda, \lambda)\,\mathrm{d}\lambda \\
&= \frac{\alpha_a}{\alpha_b}\frac{\sigma_b^2}{\sigma_a^2}\mathbf{x}_a + \int_{\lambda_a}^{\lambda_b} \frac{\alpha_\lambda}{\alpha_b}\frac{\sigma_b^2}{\sigma_\lambda^2}\left(2\,\alpha_\lambda\mathbf{D}_\theta(\mathbf{x}_\lambda) + \sqrt{2}\,\sigma_\lambda\frac{\mathrm{d}\mathbf{w}_\lambda}{\mathrm{d}\lambda}\right)\mathrm{d}\lambda \\
&= \frac{\alpha_a}{\alpha_b}\frac{\sigma_b^2}{\sigma_a^2}\mathbf{x}_a + 2\frac{\sigma_b^2}{\alpha_b}\int_{\lambda_a}^{\lambda_b}\frac{\alpha_\lambda^2}{\sigma_\lambda^2}\mathbf{D}_\theta(\mathbf{x}_\lambda)\,\mathrm{d}\lambda + \sqrt{2}\frac{\sigma_b^2}{\alpha_b}\int_{\lambda_a}^{\lambda_b}\frac{\alpha_\lambda}{\sigma_\lambda}\mathrm{d}\mathbf{w}_\lambda \\
&= \frac{\alpha_a}{\alpha_b}\frac{\sigma_b^2}{\sigma_a^2}\mathbf{x}_a + 2\,\alpha_b\,e^{-2\lambda_b}\underbrace{\int_{\lambda_a}^{\lambda_b}e^{2\lambda}\mathbf{D}_\theta(\mathbf{x}_\lambda)\,\mathrm{d}\lambda}_{\text{Use Taylor Expansion}} + \sqrt{2}\,\alpha_b\,e^{-2\lambda_b}\underbrace{\int_{\lambda_a}^{\lambda_b}e^\lambda\,\mathrm{d}\mathbf{w}_\lambda}_{\text{Itô Integral}}.
\end{aligned} \tag{13}$$

The integral $\int_{\lambda_a}^{\lambda_b}e^{2\lambda}\mathbf{D}_\theta(\mathbf{x}_\lambda)\,\mathrm{d}\lambda$ can be computed by performing Taylor Expansion:

$$\int_{\lambda_a}^{\lambda_b}e^{2\lambda}\mathbf{D}_\theta(\mathbf{x}_\lambda)\,\mathrm{d}\lambda \approx \sum_{n=0}^{k-1}\mathbf{D}_\theta^{(n)}(\mathbf{x}_{\lambda_a})\int_{\lambda_a}^{\lambda_b}e^{2\lambda}\frac{(\lambda - \lambda_a)^n}{n!}\,\mathrm{d}\lambda + \mathcal{O}((\lambda_b - \lambda_a)^{k+1}), \tag{14}$$

where $k \geq 1$, and $\mathbf{D}_\theta^{(n)}(\mathbf{x}_{\lambda_a}) := \frac{\mathrm{d}^n\mathbf{D}_\theta(\mathbf{x}_{\lambda_a})}{\mathrm{d}\lambda^n}$ is the $n^{\text{th}}$-order derivative of $\mathbf{D}_\theta(\cdot)$ w.r.t. $\lambda$.

Furthermore, we can compute the Itô integral (Rogers & Williams, 2000) as:

$$\int_{\lambda_a}^{\lambda_b}e^\lambda\,\mathrm{d}\mathbf{w}_\lambda = \left(\sqrt{\int_{\lambda_a}^{\lambda_b}e^{2\lambda}\,\mathrm{d}\lambda}\right)\mathbf{z}_b = \frac{e^{\lambda_b}}{\sqrt{2}}\sqrt{1 - e^{2(\lambda_a - \lambda_b)}}\,\mathbf{z}_b,$$

where $\mathbf{z}_b \sim \mathcal{N}(\mathbf{0}, \mathbf{I})$.

Substituting $k = 1$ for Equation (14), we can ultimately simplify Equation (13) as:

$$\begin{aligned}
\mathbf{x}_b &= \frac{\mathrm{SNR}_a}{\mathrm{SNR}_b}\frac{\alpha_b}{\alpha_a}e^{2(\lambda_a - \lambda_b)}\mathbf{x}_a + \alpha_b\left(1 - e^{2(\lambda_a - \lambda_b)}\right)\mathbf{D}_\theta(\mathbf{x}_{\lambda_a}) + \sigma_b\sqrt{1 - e^{2(\lambda_a - \lambda_b)}}\,\mathbf{z}_b \\
&= \frac{\alpha_b}{\alpha_a}\mathbf{x}_a + \alpha_b\left(1 - \frac{\mathrm{SNR}_a}{\mathrm{SNR}_b}\right)\mathbf{D}_\theta(\mathbf{x}_{\lambda_a}) + \sigma_b\sqrt{1 - \frac{\mathrm{SNR}_a}{\mathrm{SNR}_b}}\,\mathbf{z}_b,
\end{aligned} \tag{15}$$

where $\mathrm{SNR}_t := \alpha_t^2/\sigma_t^2 = e^{2\lambda_t}$.

Thus, given the initial value $\mathbf{x}_a$, where $b = a + \Delta t$ and $0 \leq b < a \leq T$, the solution to $\mathbf{x}_b$ is:

$$\mathbf{x}_b = \frac{\mathrm{SNR}_a}{\mathrm{SNR}_b}\frac{\alpha_b}{\alpha_a}\mathbf{x}_a + \alpha_b\left(1 - \frac{\mathrm{SNR}_a}{\mathrm{SNR}_b}\right)\mathbf{D}_\theta(\mathbf{x}_a) + \sigma_b\sqrt{1 - \frac{\mathrm{SNR}_a}{\mathrm{SNR}_b}}\,\mathbf{z}_b. \tag{16}$$

$\blacksquare$

## B.2 PROOF OF PROPOSITION 2

Given a well-trained DBM $\boldsymbol{D}_\theta(\cdot)$ that approximates data sample $\mathbf{x}_0$, we can re-write the Bridge PF ODE (Equation (5)) as:

$$
\frac{\mathrm{d}\mathbf{x}_t}{\mathrm{d}t} = \boldsymbol{f}(\mathbf{x}_t, t) - g(t)^2 \left( \frac{1}{2}\nabla_{\mathbf{x}_t} \log p_t(\mathbf{x}_t \mid \mathbf{x}_T) - \nabla_{\mathbf{x}_t} \log p_t(\mathbf{x}_T \mid \mathbf{x}_t) \right)
$$

$$
= \mathbf{x}_t \frac{\mathrm{d}\log\alpha_t}{\mathrm{d}t} + 2\sigma_t^2 \frac{\mathrm{d}\log\left(\frac{\alpha_t}{\sigma_t}\right)}{\mathrm{d}t} \left( \frac{1}{2}\nabla_{\mathbf{x}_t} \log p_t(\mathbf{x}_t \mid \mathbf{x}_T) - \nabla_{\mathbf{x}_t} \log p_t(\mathbf{x}_T \mid \mathbf{x}_t) \right)
$$

$$
= \mathbf{x}_t \frac{\mathrm{d}\log\alpha_t}{\mathrm{d}t} + \sigma_t^2 \frac{\mathrm{d}\log\left(\frac{\alpha_t}{\sigma_t}\right)}{\mathrm{d}t} \left[ \frac{\frac{\alpha_T^2\sigma_t^2}{\sigma_T^2\alpha_t^2}\frac{\alpha_t}{\alpha_T}\mathbf{x}_T + \alpha_t\left(1 - \frac{\alpha_T^2\sigma_t^2}{\sigma_T^2\alpha_t^2}\right)\boldsymbol{D}_\theta(\mathbf{x}_t) - \mathbf{x}_t}{\sigma_t^2\left(1 - \frac{\alpha_T^2\sigma_t^2}{\sigma_T^2\alpha_t^2}\right)} - 2\frac{\frac{\alpha_t}{\alpha_T}\mathbf{x}_T - \mathbf{x}_t}{\sigma_t^2\left(\frac{\alpha_t^2\sigma_T^2}{\sigma_t^2\alpha_T^2} - 1\right)} \right]
$$

$$
= \mathbf{x}_t \frac{\mathrm{d}\log\alpha_t}{\mathrm{d}t} + \frac{\mathrm{d}\log\left(\frac{\alpha_t}{\sigma_t}\right)}{\mathrm{d}t} \left[ \alpha_t\boldsymbol{D}_\theta(\mathbf{x}_t) - \mathbf{x}_t - \frac{\frac{\alpha_t}{\alpha_T}\mathbf{x}_T - \mathbf{x}_t}{\frac{\alpha_t^2\sigma_T^2}{\sigma_t^2\alpha_T^2} - 1} \right]
$$

$$
= \mathbf{x}_t \left( \frac{\mathrm{d}\log\alpha_t}{\mathrm{d}t} - \frac{\mathrm{d}\log\left(\frac{\alpha_t}{\sigma_t}\right)}{\mathrm{d}t} + \frac{1}{\frac{\alpha_t^2\sigma_T^2}{\sigma_t^2\alpha_T^2} - 1}\frac{\mathrm{d}\log\left(\frac{\alpha_t}{\sigma_t}\right)}{\mathrm{d}t} \right) + \alpha_t \frac{\mathrm{d}\log\left(\frac{\alpha_t}{\sigma_t}\right)}{\mathrm{d}t} \left[ \boldsymbol{D}_\theta(\mathbf{x}_t) - \frac{\mathbf{x}_T/\alpha_T}{\frac{\alpha_t^2\sigma_T^2}{\sigma_t^2\alpha_T^2} - 1} \right].
$$

We further simplify the equation above as:

$$
\frac{\mathrm{d}\mathbf{x}_t}{\mathrm{d}t} = \underbrace{\mathbf{x}_t \left( \frac{\mathrm{d}\log\sigma_t}{\mathrm{d}t} + \frac{1}{\frac{\alpha_t^2\sigma_T^2}{\sigma_t^2\alpha_T^2} - 1}\frac{\mathrm{d}\log\left(\frac{\alpha_t}{\sigma_t}\right)}{\mathrm{d}t} \right)}_{L(t)\ (\text{Linear Term})} + \underbrace{\alpha_t\left[ \boldsymbol{D}_\theta(\mathbf{x}_t) - \frac{\mathbf{x}_T/\alpha_T}{\frac{\alpha_t^2\sigma_T^2}{\sigma_t^2\alpha_T^2} - 1} \right]\frac{\mathrm{d}\log\left(\frac{\alpha_t}{\sigma_t}\right)}{\mathrm{d}t}}_{N(\mathbf{x}_t, t)\ (\text{Non-linear Term})}. \quad (17)
$$

where $L(t)$ is the *linear* term, and $N(\mathbf{x}_t, t)$ is the *non-linear* term. Thus, we can clearly observe the semi-linearity of Equation (5). Similar to the derivation above, we can once again make use of the EI method to solve Equation (17):

Given an initial value $\mathbf{x}_a$ where $b = a + \Delta\tau$ and $0 < b < a < T$, we obtain the solution to $\mathbf{x}_b$ as:

$$
\mathbf{x}_b = e^{\int_a^b L(r)\,\mathrm{d}r} \mathbf{x}_a + \int_a^b e^{\int_{a+\tau}^b L(h)\,\mathrm{d}h} \cdot N(\mathbf{x}_\tau, \tau)\,\mathrm{d}\tau. \quad (18)
$$

To simplify the equation, we first integrate the linear term $L(r)$ with respect to $r$ from $a$ to $b$:

$$
\int_a^b L(r)\,\mathrm{d}r = \left[ \log\left( \frac{\alpha_r\sqrt{e^{2(\lambda_r - \lambda_T)} - 1}}{e^{2\lambda_r}} \right) \right]_a^b = \log\left( \frac{\alpha_b}{\alpha_a} e^{2(\lambda_a - \lambda_b)} \sqrt{\frac{e^{2(\lambda_b - \lambda_T)} - 1}{e^{2(\lambda_a - \lambda_T)} - 1}} \right), \quad (19)
$$

where $\lambda_t := \log\frac{\alpha_t}{\sigma_t}$ has the inverse function $t_\lambda(\cdot)$ which satisfies $t_\lambda(\lambda_t) = t$.

Next, we can use the *change-of-variables* method for $\lambda$. We denote $\alpha_\lambda := \alpha_{t_\lambda(\lambda)}$, $\mathbf{x}_\lambda := \mathbf{x}_{t_\lambda(\lambda)}$, and $N(\mathbf{x}_\lambda, \lambda) := N(\mathbf{x}_{t_\lambda(\lambda)}, t_\lambda(\lambda))$.

Substituting Equation (19) along with the value of $N(\mathbf{x}_\lambda, \lambda)$ back into Equation (18), we get:

$$
\mathbf{x}_b = \frac{\alpha_b}{\alpha_a} e^{2(\lambda_a - \lambda_b)} \sqrt{\frac{e^{2(\lambda_b - \lambda_T)} - 1}{e^{2(\lambda_a - \lambda_T)} - 1}} \, \mathbf{x}_a
$$

$$
+ \alpha_b\, e^{-2\lambda_b} \underbrace{\int_{\lambda_a}^{\lambda_b} \frac{e^{2\lambda}}{\alpha_\lambda} \sqrt{\frac{e^{2(\lambda_b - \lambda_T)} - 1}{e^{2(\lambda - \lambda_T)} - 1}} \cdot \alpha_\lambda \left[ \boldsymbol{D}_\theta(\mathbf{x}_\lambda) - \frac{\mathbf{x}_T/\alpha_T}{e^{2(\lambda - \lambda_T)} - 1} \right]\mathrm{d}\lambda}_{\text{Separate and simplify integral further}} \quad (20)
$$

When we simplify the integral in Equation (20), we get:

$$\sqrt{e^{2(\lambda_b - \lambda_T)} - 1} \int_{\lambda_a}^{\lambda_b} \left[ \frac{e^{2\lambda} \, \boldsymbol{D_\theta}(\mathbf{x}_\lambda)}{\sqrt{e^{2(\lambda - \lambda_T)} - 1}} - \frac{\mathbf{x}_T \, e^{2\lambda}}{\alpha_T \left( e^{2(\lambda - \lambda_T)} - 1 \right)^{3/2}} \right] d\lambda. \tag{21}$$

Next, we separately integrate the individual terms of Equation (21).

For the integral associated with $\mathbf{x}_T$, we use *integration-by-parts* to simplify it:

$$\int_{\lambda_a}^{\lambda_b} \frac{\mathbf{x}_T \, e^{2\lambda}}{\alpha_T \left( e^{2(\lambda - \lambda_T)} - 1 \right)^{3/2}} \, d\lambda = -\frac{\mathbf{x}_T}{\alpha_T} \frac{e^{2\lambda_T}}{\sqrt{e^{2(\lambda_b - \lambda_T)} - 1}} \left( 1 - \sqrt{\frac{e^{2(\lambda_b - \lambda_T)} - 1}{e^{2(\lambda_a - \lambda_T)} - 1}} \right) \tag{22}$$

Using the result in Equation (22), we can simplify Equation (20) as:

$$\mathbf{x}_b = \frac{\alpha_b}{\alpha_a} e^{2(\lambda_a - \lambda_b)} \sqrt{\frac{\rho(\lambda_b, \lambda_T)}{\rho(\lambda_a, \lambda_T)}} \, \mathbf{x}_a + \frac{\alpha_b}{\alpha_T} e^{2(\lambda_T - \lambda_b)} \left( 1 - \sqrt{\frac{\rho(\lambda_b, \lambda_T)}{\rho(\lambda_a, \lambda_T)}} \right) \mathbf{x}_T$$

$$+ \alpha_b \, e^{-2\lambda_b} \sqrt{\rho(\lambda_b, \lambda_T)} \underbrace{\int_{\lambda_a}^{\lambda_b} \frac{e^{2\lambda} \, \boldsymbol{D_\theta}(\mathbf{x}_\lambda)}{\sqrt{\rho(\lambda, \lambda_T)}} \, d\lambda}_{\text{Use Taylor Expansion}}, \tag{23}$$

where $\rho(m, n) := e^{2(m-n)} - 1$.

Finally, we perform Taylor Expansion to obtain the solution to the integral in Equation (23):

$$\int_{\lambda_a}^{\lambda_b} \frac{e^{2\lambda} \, \boldsymbol{D_\theta}(\mathbf{x}_\lambda)}{\sqrt{\rho(\lambda, \lambda_T)}} \, d\lambda \approx \sum_{n=0}^{k-1} \underbrace{\boldsymbol{D_\theta}^{(n)}(\mathbf{x}_{\lambda_a})}_{\text{Estimated}} \underbrace{\int_{\lambda_a}^{\lambda_b} \frac{e^{2\lambda}}{\sqrt{\rho(\lambda, \lambda_T)}} \frac{(\lambda - \lambda_a)^n}{n!} d\lambda}_{\text{Analytically Computed (Section B.3)}} + \underbrace{\mathcal{O}((\lambda_b - \lambda_a)^{k+1})}_{\text{Omitted}},$$

where $k \geq 1$, and $\boldsymbol{D_\theta}^{(n)}(\mathbf{x}_{\lambda_a}) := \frac{d^n \boldsymbol{D_\theta}(\mathbf{x}_{\lambda_a})}{d\lambda^n}$ is the $n^{\text{th}}$-order derivative of $\boldsymbol{D_\theta}(\cdot)$ w.r.t. $\lambda$. This is the same as Equation (8).

Thus, we can derive an exact solution for $\mathbf{x}_b$. For completeness, we derive the $1^{\text{st}}$- and $2^{\text{nd}}$-order solutions below.

∎

### B.3 Deriving Solutions for Proposition 2

### B.4 $1^{\text{ST}}$-order Solution

We use Taylor Expansion to find the solution. By using $k = 1$, the $1^{\text{st}}$-order solution is as follows:

$$\int_{\lambda_a}^{\lambda_b} \frac{e^{2\lambda} \, \boldsymbol{D_\theta}(\mathbf{x}_\lambda)}{\sqrt{\rho(\lambda, \lambda_T)}} \, d\lambda \approx \boldsymbol{D_\theta}^{(0)}(\mathbf{x}_{\lambda_a}) \int_{\lambda_a}^{\lambda_b} \frac{e^{2\lambda}}{\sqrt{\rho(\lambda, \lambda_T)}} \frac{(\lambda - \lambda_a)^0}{0!} d\lambda$$

$$= \boldsymbol{D_\theta}(\mathbf{x}_{\lambda_a}) \int_{\lambda_a}^{\lambda_b} \frac{e^{2\lambda}}{\sqrt{\rho(\lambda, \lambda_T)}} \, d\lambda$$

$$= \boldsymbol{D_\theta}(\mathbf{x}_{\lambda_a}) \, e^{2\lambda_T} \sqrt{\rho(\lambda_b, \lambda_T)} \left( 1 - \sqrt{\frac{\rho(\lambda_a, \lambda_T)}{\rho(\lambda_b, \lambda_T)}} \right). \tag{24}$$

Substituting Equation (24) back into Equation (23), we get the following $1^{\text{st}}$-order formulation of $\mathbf{x}_b$:

$$\mathbf{x}_b = \frac{\alpha_b}{\alpha_a} e^{2(\lambda_a - \lambda_b)} \sqrt{\frac{\rho(\lambda_b, \lambda_T)}{\rho(\lambda_a, \lambda_T)}} \, \mathbf{x}_a + \frac{\alpha_b}{\alpha_T} e^{2(\lambda_T - \lambda_b)} \left( 1 - \sqrt{\frac{\rho(\lambda_b, \lambda_T)}{\rho(\lambda_a, \lambda_T)}} \right) \mathbf{x}_T$$

$$+ \alpha_b \, e^{2(\lambda_T - \lambda_b)} \rho(\lambda_b, \lambda_T) \left( 1 - \sqrt{\frac{\rho(\lambda_a, \lambda_T)}{\rho(\lambda_b, \lambda_T)}} \right) \boldsymbol{D_\theta}(\mathbf{x}_{\lambda_a}). \tag{25}$$

∎

**Relation to DBIM Sampler (Zheng et al., 2024).** Our solution is a generalized form of the DBIM Sampler. Simplifying the DBIM's formulation for $\mathbf{x}_b$ (with $\rho = 0$), we get:

$$
\mathbf{x}_b = \frac{\alpha_b}{\alpha_T} \frac{\mathrm{SNR}_T}{\mathrm{SNR}_b} \mathbf{x}_T + \alpha_b \left( 1 - \frac{\mathrm{SNR}_T}{\mathrm{SNR}_b} \right) \boldsymbol{D}_\theta(\mathbf{x}_a)
$$

$$
+ \sigma_b \sqrt{1 - \frac{\mathrm{SNR}_T}{\mathrm{SNR}_b}} \left[ \frac{\mathbf{x}_a - \frac{\alpha_a}{\alpha_T} \frac{\mathrm{SNR}_T}{\mathrm{SNR}_a} \mathbf{x}_T - \alpha_a \left( 1 - \frac{\mathrm{SNR}_T}{\mathrm{SNR}_a} \right) \boldsymbol{D}_\theta(\mathbf{x}_a)}{\sigma_a \sqrt{1 - \frac{\mathrm{SNR}_T}{\mathrm{SNR}_a}}} \right]
$$

$$
= \frac{\alpha_b}{\alpha_a} \frac{\mathrm{SNR}_a}{\mathrm{SNR}_b} \sqrt{\frac{\frac{\mathrm{SNR}_b}{\mathrm{SNR}_T} - 1}{\frac{\mathrm{SNR}_a}{\mathrm{SNR}_T} - 1}} \mathbf{x}_a + \frac{\alpha_b}{\alpha_T} \frac{\mathrm{SNR}_T}{\mathrm{SNR}_b} \left( 1 - \frac{\sigma_b \sqrt{1 - \frac{\mathrm{SNR}_T}{\mathrm{SNR}_b}}}{\sigma_a \sqrt{1 - \frac{\mathrm{SNR}_T}{\mathrm{SNR}_a}}} \frac{\alpha_a}{\alpha_T} \frac{\mathrm{SNR}_T}{\mathrm{SNR}_a} \frac{\alpha_T}{\alpha_b} \frac{\mathrm{SNR}_b}{\mathrm{SNR}_T} \right) \mathbf{x}_T
$$

$$
+ \alpha_b \frac{\mathrm{SNR}_T}{\mathrm{SNR}_b} \left( \frac{\mathrm{SNR}_b}{\mathrm{SNR}_T} - 1 \right) \left( 1 - \sqrt{\frac{\frac{\mathrm{SNR}_a}{\mathrm{SNR}_T} - 1}{\frac{\mathrm{SNR}_b}{\mathrm{SNR}_T} - 1}} \right) \boldsymbol{D}_\theta(\mathbf{x}_a). \tag{26}
$$

By substituting the equation above with $e^{2\lambda_t} := \mathrm{SNR}_t$, we in fact see that Equation (26) simplifies to our 1$^{\text{st}}$-order formulation of $\mathbf{x}_b$ in Equation (25). Thus, we see that DBIM is actually a 1$^{\text{st}}$-order formulation of our solution to the Bridge PF ODE (*i.e.*, $k = 1$). DBMSolver's advantage is that it instead utilizes a more precise, 2$^{\text{nd}}$-order solution that has lower error bounds compared to DBIM.

## B.5 2$^{\text{ND}}$-ORDER SOLUTION

Similar to the derivation in Section B.4, we use Taylor Expansion to find the solution when $k = 2$:

$$
\int_{\lambda_a}^{\lambda_b} \frac{e^{2\lambda} \boldsymbol{D}_\theta(\mathbf{x}_\lambda)}{\sqrt{\rho(\lambda, \lambda_T)}} \, \mathrm{d}\lambda \approx \underbrace{\boldsymbol{D}_\theta^{(0)}(\mathbf{x}_{\lambda_a}) \int_{\lambda_a}^{\lambda_b} \frac{e^{2\lambda}}{\sqrt{\rho(\lambda, \lambda_T)}} \, \mathrm{d}\lambda}_{\text{Solution derived in Equation (24)}} + \underbrace{\boldsymbol{D}_\theta^{(1)}(\mathbf{x}_{\lambda_a}) \int_{\lambda_a}^{\lambda_b} \frac{e^{2\lambda}(\lambda - \lambda_a)}{\sqrt{\rho(\lambda, \lambda_T)}} \, \mathrm{d}\lambda}_{\text{Solution derived below}}. \tag{27}
$$

The second term's integral can be solved as:

$$
\int_{\lambda_a}^{\lambda_b} \frac{e^{2\lambda}(\lambda - \lambda_a)}{\sqrt{\rho(\lambda, \lambda_T)}} \, \mathrm{d}\lambda = e^{2\lambda_T} \left[ \tan^{-1} \left( \sqrt{\rho(\lambda_b, \lambda_T)} \right) - \tan^{-1} \left( \sqrt{\rho(\lambda_a, \lambda_T)} \right) \right]
$$

$$
+ e^{2\lambda_T} \left[ (\lambda_b - \lambda_a - 1) \sqrt{\rho(\lambda_b, \lambda_T)} + \sqrt{\rho(\lambda_a, \lambda_T)} \right]. \tag{28}
$$

By substituting Equations 24 and 28 into Equation (27), we reach the 2$^{\text{nd}}$-order Taylor Expansion:

$$
e^{2\lambda_T} \sqrt{\rho(\lambda_b, \lambda_T)} \left( 1 - \sqrt{\frac{\rho(\lambda_a, \lambda_T)}{\rho(\lambda_b, \lambda_T)}} \right) \left[ \boldsymbol{D}_\theta(\mathbf{x}_{\lambda_a}) - \boldsymbol{D}_\theta^{(1)}(\mathbf{x}_{\lambda_a}) \right]
$$

$$
+ e^{2\lambda_T} \sqrt{\rho(\lambda_b, \lambda_T)} \left[ \lambda_b - \lambda_a + \frac{\tan^{-1} \left( \sqrt{\rho(\lambda_b, \lambda_T)} \right) - \tan^{-1} \left( \sqrt{\rho(\lambda_a, \lambda_T)} \right)}{\sqrt{\rho(\lambda_b, \lambda_T)}} \right] \boldsymbol{D}_\theta^{(1)}(\mathbf{x}_{\lambda_a}). \tag{29}
$$

Finally, substituting Equation (29) back into Equation (23), we get the following formulation of $\mathbf{x}_b$:

$$
\mathbf{x}_b = \frac{\alpha_b}{\alpha_a} e^{2(\lambda_a - \lambda_b)} \sqrt{\frac{\rho(\lambda_b, \lambda_T)}{\rho(\lambda_a, \lambda_T)}} \mathbf{x}_a + \frac{\alpha_b}{\alpha_T} e^{2(\lambda_T - \lambda_b)} \left( 1 - \sqrt{\frac{\rho(\lambda_b, \lambda_T)}{\rho(\lambda_a, \lambda_T)}} \right) \mathbf{x}_T
$$

$$
- \alpha_b \, \rho(\lambda_T, \lambda_b) \left[ 1 - \sqrt{\frac{\rho(\lambda_a, \lambda_T)}{\rho(\lambda_b, \lambda_T)}} \right] \left( \boldsymbol{D}_\theta(\mathbf{x}_{\lambda_a}) - \boldsymbol{D}_\theta^{(1)}(\mathbf{x}_{\lambda_a}) \right)
$$

$$
- \alpha_b \, \rho(\lambda_T, \lambda_b) \left[ \lambda_b - \lambda_a + \frac{\tan^{-1} \left( \sqrt{\rho(\lambda_b, \lambda_T)} \right) - \tan^{-1} \left( \sqrt{\rho(\lambda_a, \lambda_T)} \right)}{\sqrt{\rho(\lambda_b, \lambda_T)}} \right] \boldsymbol{D}_\theta^{(1)}(\mathbf{x}_{\lambda_a}), \tag{30}
$$

where $D_\theta^{(1)}(\mathbf{x}_{\lambda_a}) \approx \frac{D_\theta(\tilde{\mathbf{x}}_{\lambda_m}) - D_\theta(\mathbf{x}_{\lambda_a})}{\lambda_m - \lambda_a}$, with $\lambda_m := (1 - r)\lambda_a + r\lambda_b$ and a ratio hyperparameter $r \in [0, 1]$. Note that $\tilde{\mathbf{x}}_{\lambda_m}$, the perturbed image at $\lambda_m$, is obtained via Equation (25).

∎

## C  EXPERIMENT DETAILS

### C.1  TRAINING DETAILS

We provide thorough details for the diffusion bridge models and their training procedures in Table 7.

Table 7: Training details for the various Image-to-Image Translation tasks.

| Dataset | Edges2Handbags [14] | DIODE [40] | Inpainting on Conditional ImageNet [2] | CelebAMask-HQ [19] | Face2Comics [39] |
|---|---|---|---|---|---|
| **Hyperparameters and Training Details** | | | | | |
| Bridge Formulation | VP | VP | I²SB [22] | VP | VP |
| Noise Conditioning, $c_{\text{noise}}$ | $250 \ln t$ | $250 \ln t$ | $1000\,t$ | $250 \ln t$ | $250 \ln t$ |
| Learning Rate | 1e-4 | 1e-4 | 1e-4 | 2e-4 | 2e-4 |
| EMA Rate | 0.9999 | 0.9999 | 0.9999 | 0.9993 | 0.9993 |
| Noise Discretization Schedule | Karras | Karras | Karras | Karras | Karras |
| Noise Discretization Steps | 40 | 40 | 40 | 40 | 40 |
| Batch Size | 256 | 64 | 256 | 64 | 64 |
| Training Iterations | 400k | 400k | 400k | 120k | 120k |
| Number and Type of GPUs | 4 A100 | 4 A100 | 8 A800 | 8 A6000 | 8 A6000 |
| **Model Details** | | | | | |
| Model Channels | 192 | 256 | 256 | 256 | 256 |
| Dropout | 10% | 10% | 10% | 10% | 10% |
| Time Embedding | Cosine | Cosine | Cosine | Cosine | Cosine |
| Channel Multiplier | (1, 2, 3, 4) | (1, 1, 2, 2, 4, 4) | (1, 1, 2, 2, 4, 4) | (1, 1, 2, 2, 4, 4) | (1, 1, 2, 2, 4, 4) |
| Number of Residual Layers | 3 | 2 | 2 | 2 | 2 |
| Attention Resolutions | (8, 16, 32) | (8, 16, 32) | (8, 16, 32) | (8, 16, 32) | (8, 16, 32) |
| Model Capacity (Mparams) | 284 | 534 | 534 | 534 | 534 |

### C.2  SAMPLING DETAILS

Table 8: Sampling details for the various Image-to-Image Translation tasks.

| Dataset | Edges2Handbags [14] | DIODE [40] | Inpainting on Conditional ImageNet [2] | CelebAMask-HQ [19] | Face2Comics [39] |
|---|---|---|---|---|---|
| **Hyperparameters for Sampling** | | | | | |
| Discretization Schedule | Karras | Karras | Uniform | Uniform | Karras |
| Discretization Steps | 40 | 40 | 40 | 40 | 40 |

## D  MORE QUALITATIVE RESULTS

Beyond the quantitative metrics presented earlier, we include additional qualitative results in Figures 8–12 to further highlight the perceptual advantages of DBMSolver. While most baseline methods fail to produce visually coherent or structurally faithful outputs at low NFEs, only DBIM offers a somewhat competitive baseline. However, even in direct comparison, DBMSolver consistently exhibits superior fidelity, texture richness, and semantic alignment. We encourage readers to closely examine the nuanced differences in Figures 11 and 9, particularly between DBIM and our method. For instance, in the DIODE samples, DBMSolver preserves delicate edge structures and fine-grained details in tree branches and twigs that are noticeably degraded in DBIM outputs. Similarly, in the Edges2Handbags domain, our method captures subtle material textures and contour sharpness that DBIM tends to blur or oversimplify. These visual distinctions reinforce the efficacy of our approach and underscore its robustness across diverse generative tasks. Further inpainting results are also presented in Fig. 12.

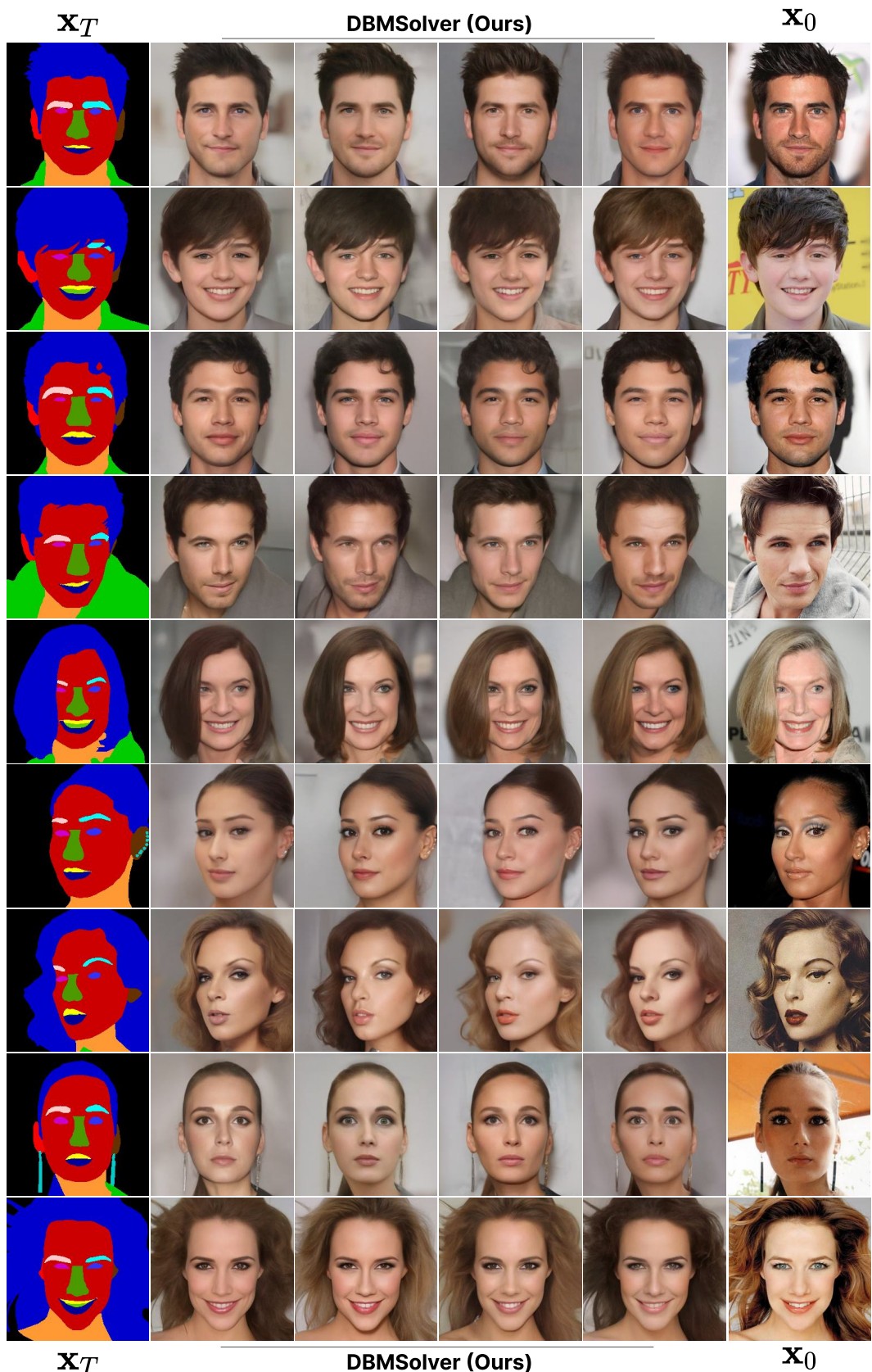

Figure 8: Additional CelebAMask-HQ samples for DBMSolver with 6 NFEs, with different initial SDE steps.

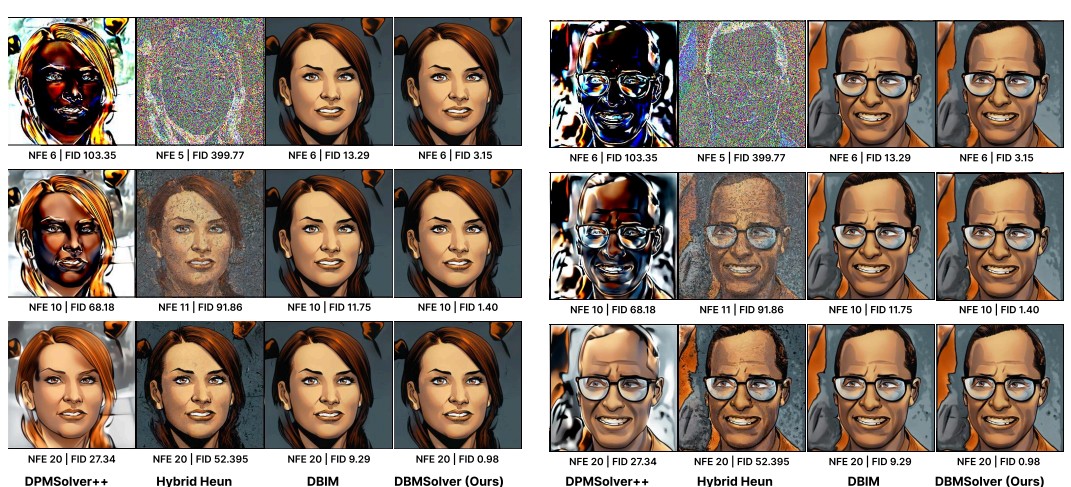

Figure 9: Additional qualitative comparison on Face2Comics.

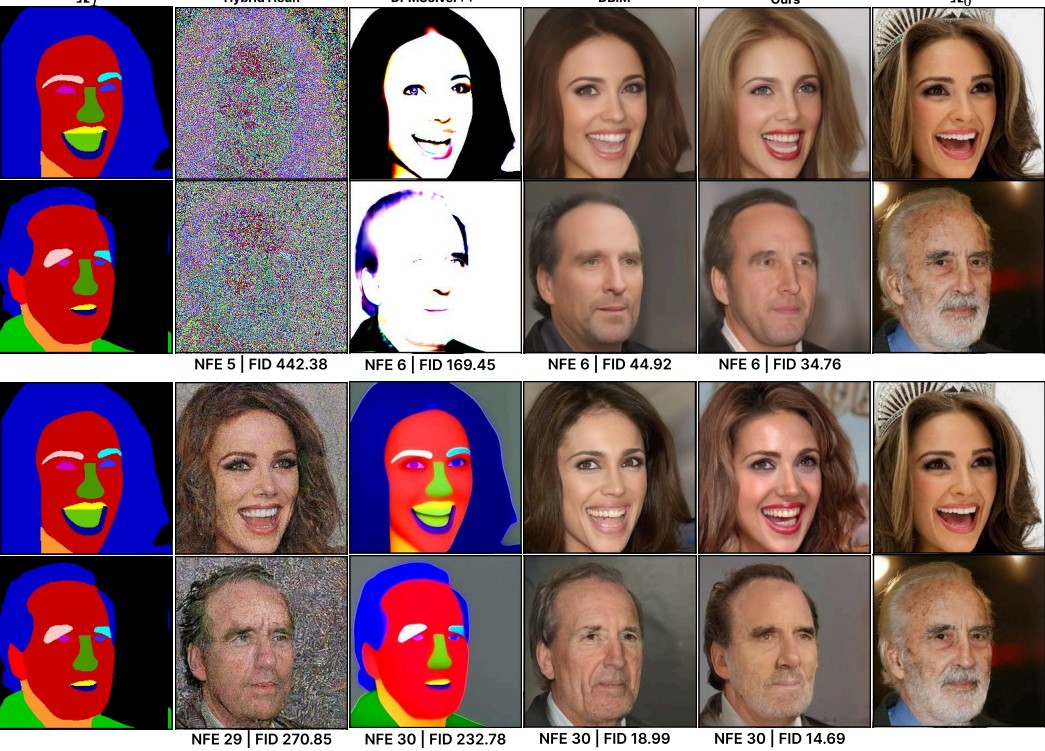

Figure 10: Additional qualitative comparison for Label-to-Face Generation on CelebAMask-HQ.

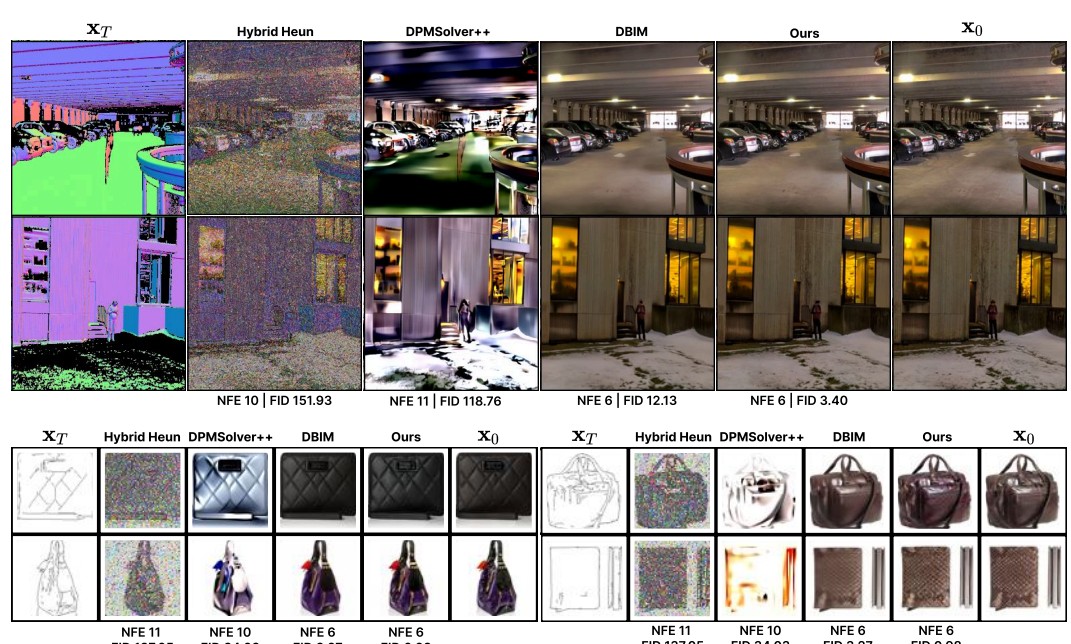

Figure 11: Additional qualitative comparison on DIODE (top) and Edges2Handbags (bottom).

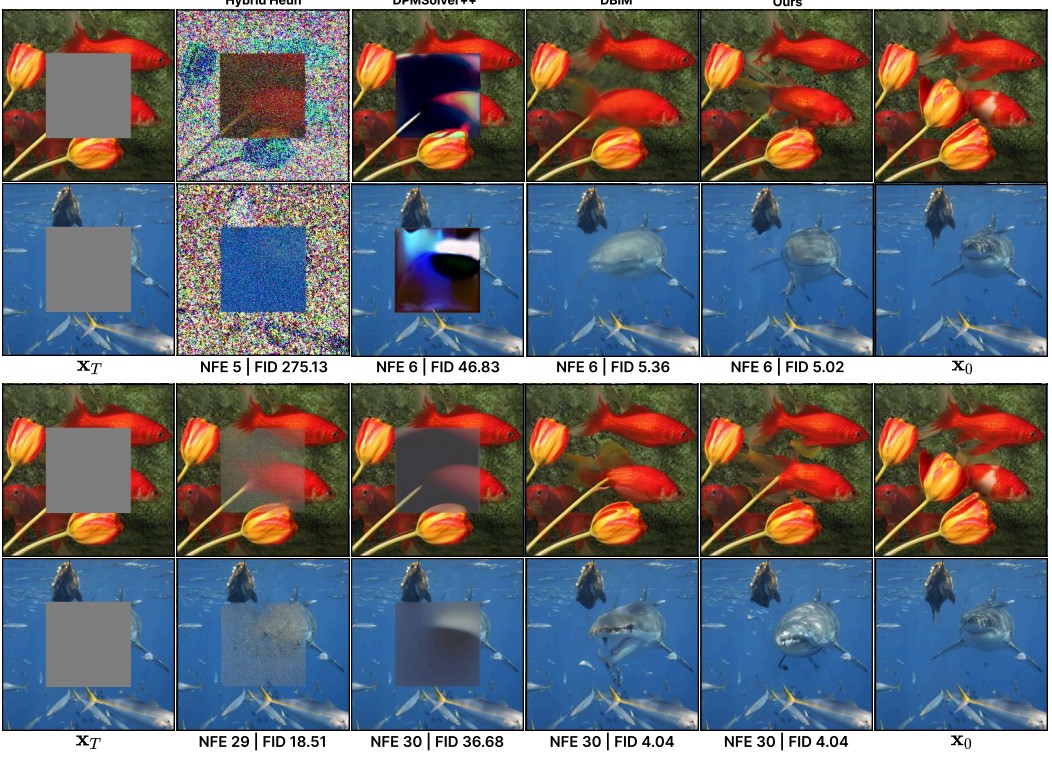

Figure 12: Additional qualitative comparison for Class-Conditional Inpainting on ImageNet.

