# OpenReview forum: "DBMSolver: Fast Diffusion Bridge Sampling for High-Quality Image-to-Image Translation"
_ICLR.cc/2026/Conference — ICLR 2026 Conference Withdrawn Submission_

### Official Review · Reviewer_8sb8 · 2025-10-15

**Soundness:** 2
**Presentation:** 3
**Contribution:** 2
**Rating:** 4
**Confidence:** 3

**Summary:**

The paper proposed a training free algorithm DBMSolver to accelerate existing diffusion bridge models for image-to-image translation problems, which is based on high order exponential integrator theory for bridge SDE and bridge ODE. The theoretical contributions include Proposition 1 and Proposition 2, which give closed formulas for solutions of bridge SDE and bridge ODE, respectively. The evaluation of the proposed DBMSolver includes sketch-to-image on edges2handbags, normals-to-image on DIODE-Outdoor, face-to-comic, image inpainting with central masks on ImageNet, and semantic label-to-face on CelebAMask-HQ. The method is compared with closely related exponential integrator methods DBIM, DDBM, and DPM-Solver++ using image realism quality metric FID and inference time. The results of the DBMSolver show its superiority against compared methods in terms of FID while maintaining small NFE between 6 and 30.

**Strengths:**

1. Proposition 1 and 2 provide general results for solutions of Bridge SDE and Bridge ODE, which can be applicable for other data-to-data bridge models, for example, audio restoration models and text-to-speech models.
2. Figure 2 shows faster convergence in terms of FID for DBMSolver with the increase of NFE compared with DBIM, so 6-30 NFE is enough for all considered image-to-image translation problems with DBMSolver while being a training free method.
3. The method alleviates the problem of divergence of Bridge ODE for the initial step from Proposition 2 using the Bridge SDE solution from Proposition 1, which unifies the proposed theoretical results.
4. The DBMSolver was tested on 5 various image-to-image translation problems, which supports the practical usage of the method.

**Weaknesses:**

1. Lack of comparisons. I appreciate that the major advantage of the proposed method is that its training free, which is different from other methods for acceleration of diffusion bridge models, such as distillation. However, there is a gap between NFE, which is used in DBMSolver, and distillation method, such as CDBM (He et al.) and IBMD (Gushchin et al.), which achieved good I2I results with 2 or 1 NFE. In particular, Table 5 and 6 in IBMD show that the distillation models achieve very close results to the results, which are reported in Table 2 and 5 in DBMSolver with NFE=6, while using only NFE=2 or NFE=1. The paper lacks of clear discussion of practical results of the proposed training free method in relation to existing fast distillation diffusion bridge models.
2. Lack of evaluation. The method used only the FID metric to evaluate the image generation quality for 4 image-to-image translation problems: DIODE, edges-to-handbags, face-to-comics, CelebAMask-HQ. In DBIM and DDBM methods, diffusion bridge models for the problems of DIODE and edges-to-handbags were evaluated with other metrics - paired LPIPS and MSE, and image generation IS. Additional image quality metrics (paired or no-reference) are important in the light of the observation in Section 5.2 of IBMD work, where the authors found that for edges-to-handbags and DIODE-Outdoor image-to-image translation problems the evaluation protocol of DDBM and DBIM leads to report of FID on training set, while testing sets are too small for FID computation.
3. Lack of discussion of existing theoretical results regarding exponential integrator for bridge diffusion models with the proposed Proposition 1 and 2. The novelty of the proposed theoretical results should be highlighted in relation to existing methods, which apply exponential integrator theory for data-to-data translation models with diffusion bridges. In particular, the results of Proposition 1 and 2 in DBMSolver seem to be very close to the results of Proposition 3.2 in Bridge-TTS work of Chen et al.
4. Lack of ablation studies. Since the DBMSolver mixes steps from Bridge ODE and Bridge SDE solutions, there is a question which formulation provides better practical results. In particular, authors used $k = 2$ for Bridge ODE, but didn't explore the option $k = 2$ for Bridge SDE. I also expect that Bridge SDE formulation provides a diversity, which is important for multimodal image-to-image translation problems (see also Table 1 and Table 2 in BBDM). The explanation behind the choice of Bridge ODE instead of Bridge SDE would explain the method better.

References:

CDBM - Consistency Diffusion Bridge Models. NeurIPS-2024.

IBMD - Inverse Bridge Matching Distillation. ICML-2025.

Bridge-TTS - Schrodinger Bridges Beat Diffusion Models on Text-to-Speech Synthesis, arxiv, 2023 (https://arxiv.org/abs/2312.03491v1).

BBDM - BBDM: Image-to-image Translation with Brownian Bridge Diffusion Models, CVPR-2023.

**Questions:**

1) Can you comment on comparison between DBMSolver and fast distillation bridge diffusion models for image-to-image translation, such as CDBM and IBDM in terms of quality and inference speed?
2) Can you provide other metrics to evaluate DBMSolver on image-to-image translation problems, such as LPIPS, MSE and IS?
3) Can you explain the choice behind Bridge ODE and Bridge SDE in your method? Can DBMSolver be applicable with Bridge SDE formulation and $k = 2$?
4) Can you comment on diversity of DBMSolver?
5) Since there is an issue with the evaluation protocol for DIODE-Outdoor and edge-to-handbags problems, can you comment the results of your method on other image-to-image translation problems, like JPEG image restoration, which was considered in I2SB (Table 3) and IBMD (Table 2 and Table 4) methods with bridge diffusion models? My question about these problems is because there is a gap between the results of teacher and student diffusion bridge models, as shown in Table 2 and Table 4 of IBMD.
6) Can you comment on the relation of Proposition 1 and 2 in DBMSolver and Proposition 3.2 in Bridge-TTS.

References:
I2SB: I2SB: Image-to-Image Schrodinger Bridge, ICML-2023.
Bridge-TTS - Schrodinger Bridges Beat Diffusion Models on Text-to-Speech Synthesis, arxiv, 2023 (https://arxiv.org/abs/2312.03491v1).

---

### Official Review · Reviewer_UUM2 · 2025-10-17

**Soundness:** 3
**Presentation:** 3
**Contribution:** 1
**Rating:** 2
**Confidence:** 3

**Summary:**

This paper presents DBMSolver, a training-free sampler for Diffusion Bridge Models that uses their semi-linear structure and exponential integrators to sample faster with fewer function evaluations.

**Strengths:**

(i) Empirically, the method attains the strongest NFE vs performance trade off in terms of the standard I2I metrics among training-free baselines.

(ii) The evaluation covers a broader set of domains than prior work DDBM [1] and DBIM [2], specifically label to face generation on CelebA and image stylization on Face2Comics.

[1] https://arxiv.org/abs/2309.16948 Denoising Diffusion Bridge Models

[2] https://arxiv.org/abs/2405.15885 Diffusion Bridge Implicit Models

**Weaknesses:**

(i) The main idea of the paper (Proposition 2) seems to already exist in [2] (see Appendix C.4, Eq. 60) and was tested there (Table 6). The only difference between the proposed Algorithm 1 and the earlier DBIM (high-order) algorithm in [2] that I see is the final Euler update step. If this is true, then the claimed novelty is overstated. The authors should clearly explain what is new compared to [2].

(ii) There are also some problems with how the theory is presented:

   1. Proposition 1: The authors claim to give an exact solution for the SDE from $x_s$ to $x_t$, but the proof (line 744) uses a first-order Taylor approximation (k = 1). So, it is not exact, and this should be made clear in the text.

   2. Incorrect score functions in Equations (2) and (5): These equations use $\nabla_{x_t} \log p_t(x_t \mid x_0, x_T)$, but based on DDBM [1] (see Theorem 1), they should use $\nabla_{x_t} \log q_t(x_t \mid x_T)$. These two are related as:$$q_t(x_t \mid x_T) = \int p_t(x_t \mid x_0, x_T) q_{\text{data}}(x_0\mid x_T) dx_0$$ (see [1], Appendix A.3).

**Questions:**

(i) Can the authors clearly explain the theoretical difference between their method and the high-order DBIM sampling in [2]? This would help to understand what is new in this work.

(ii) If the methods are different, it would be more fair to include a direct comparison with high-order DBIM in the experiments.

(iii) Based on the comments above, I suggest the authors revise Proposition 1 and Equations (2) and (5) to be more accurate.

---

### Official Review · Reviewer_3DfM · 2025-10-29

**Soundness:** 3
**Presentation:** 4
**Contribution:** 4
**Rating:** 6
**Confidence:** 4

**Summary:**

This work identifies a critical limitation in existing Diffusion Bridge Models (DBMs): current samplers are primarily based on stochastic differential equations (SDEs), which introduce two major issues: low sampling efficiency requiring a large number of steps, and stochasticity from random noise at each step, leading to output uncertainty. While these challenges have been recognized in score-based diffusion probabilistic models (DPMs), they remain unresolved in DBMs. In this study, the authors observe that the Bridge ODE retains the semi-linear structure of the Bridge SDE while enabling fast convergence through Taylor expansion. Building on this insight, they propose DBMSolver, a novel ODE-based diffusion bridge sampler that eliminates the need for stochastic sampling while maintaining high fidelity. Extensive experiments demonstrate that DBMSolver significantly outperforms existing benchmarks in terms of FID and NFE, greatly enhancing the practicality and efficiency of DBMs in image-to-image (I2I) translation tasks.

**Strengths:**

This paper is well-written and easy to follow. DBMSolver exhibits significant advantages through its efficient and training-free sampling mechanism. By introducing a novel sampler that requires no additional training or fine-tuning, it substantially accelerates the sampling process of DBMs and applies seamlessly to both conditional and unconditional I2I translation tasks. Grounded in a rigorous theoretical foundation, DBMSolver provides exact analytical solutions to ODEs governing diffusion bridge dynamics, ensuring both mathematical soundness and interpretability. Extensive experiments across diverse I2I tasks and image resolutions demonstrate that DBMSolver consistently achieves state-of-the-art performance, outperforming prior leading methods in both image quality and computational efficiency.

**Weaknesses:**

1. The paper does not provide a code example or supplementary materials related to implementation. Including a code link or additional details about the experimental setup would greatly enhance reproducibility.

2. The current experiments are conducted on relatively simple datasets and tasks. It would strengthen the work to evaluate the proposed method on more complex scenarios, such as image editing, to further demonstrate its generalization and robustness.

**Questions:**

1. In Section 3.3, the paper mentions that an SDE solver is required for the initial steps due to $\rho(\lambda_s, \lambda_T) = 0$. Could the same issue not be addressed by introducing a small constant in the denominator instead? This alternative seems more intuitive and might simplify the implementation.

2. In Appendix B.2, it is unclear why the derivative of the simple logSNR term $\frac{d\log{\frac{\alpha_t}{\sigma_t}}}{dt}$ is omitted, while the derivative of the more complex logSNR term $\frac{1}{\frac{SNR_t}{SNR_T} - 1}\frac{d\log{\frac{\alpha_t}{\sigma_t}}}{dt}$ is retained. Intuitively, discarding the complex term could lead to a more concise and elegant derivation.

---

### Official Review · Reviewer_gvHy · 2025-10-31

**Soundness:** 2
**Presentation:** 3
**Contribution:** 2
**Rating:** 4
**Confidence:** 4

**Summary:**

The paper proposes DBMSolver, a training-free fast sampler for Diffusion Bridge Models (DBMs) used in image-to-image translation. It derives (i) a closed-form step for the bridge SDE and (ii) an exponentially-integral form for the bridge probability-flow ODE, then uses a truncated Taylor expansion ($k=2$) to build a few-step, higher-order sampler. Experiments on E2H, DIODE, Face2Comics, ImageNet inpainting, and CelebAMask-HQ report better FID at low NFE than baselines.

**Strengths:**

– The paper is mostly well written.

– The method is training free, yielding strong results relative to low NFE without any fine-tuning or retraining procedure.

**Weaknesses:**

– Novelty is incremental relative to DPM-Solver. The work largely transposes DPM-Solver’s semi-linear/exponential-integrator recipe to bridges. With closed-form DBM SDE/ODE from [1] and the generalized ODE solution from [2], the framework appears to extend by swapping in bridge formulas. Please state precisely what is technically new beyond adapting DPM-Solver to DBMs and the initial SDE step.

– The bridge-SDE derivation itself does not seem to add practical benefit beyond the standard one SDE “warm” step used to avoid singularity (as in DBIM [3]) before switching to ODE sampling; Prop. 1 and the algorithm that sampling SDE style at the first step the look very close to DBIM’s ODE sampler.

– Appendix B.3 is missing. Therefore, I couldn't check on the analytical solution of Exponential Integral.

– DPM-Solver originally provides $k \in \\{ 1,2,3 \\} $  with order proofs; here the final algorithm fixes $k=2$. A higher-order ablation should be included to justify the chosen order.

– DBMSolver’s gains over DBIM are marginal in several settings. E.g., ImageNet-256 @ NFE=20: DBIM (in original paper) **4.07** vs DBMSolver **4.07**; @ NFE=10 with 2nd-order DBIM vs 2nd-order DBMSolver: **4.33** vs **4.38**. Please compare against DBIM-2nd-order at matched NFE and settings for fairness.

– Result trends are inconsistent on new datasets: while E2H, DIODE, and ImageNet inpainting are close between DBIM and DBMSolver, CelebA-MaskHQ and Face2Comics show large DBMSolver gains. Please explain the cause of this discrepancy.

**Questions:**

– What is formally new beyond adapting DPM-Solver’s exponential-integrator to bridges and adding an initial SDE step? Any identity unique to bridges?

– Do you provide a convergence/order analysis for DBMSolver on bridges , analogous to DPM-Solver’s guarantees?

– Why fix $k=2$? What is the effectiveness of higher order (e.g., $k=3$)?

– Please compare DBMSolver vs DBIM (2nd-order) at the same NFE since DBMSolver use 2nd-order solver

– Why does DBMSolver outperform DBIM much more on CelebA-MaskHQ/Face2Comics than on E2H/DIODE/ImageNet?

[1] Zhou, et al. "Denoising Diffusion Bridge Models", ICLR 2024

[2] Lu, et al. “DPM-Solver: A Fast ODE Solver for Diffusion Probabilistic Model Sampling in Around 10 Steps”, NeurIPS 2022

[3] Zheng, et al. “Diffusion Bridge Implicit Models”, ICLR 2025

---

### Note · Authors · 2025-11-13

I have read and agree with the venue's withdrawal policy on behalf of myself and my co-authors.